# MedCEG: Reinforcing Verifiable Medical Reasoning with Critical Evidence Graph

## Abstract

Large language models with reasoning capabilities have demonstrated impressive performance across a wide range of domains. In clinical applications, a transparent, step-by-step reasoning process provides physicians with strong evidence to support decision-making. While reinforcement learning has effectively enhanced reasoning performance in medical contexts, the clinical reliability of these reasoning processes remains limited because their accuracy and validity are often overlooked during training. To address this gap, we propose MedCEG, a framework that augments medical models with clinically valid reasoning pathways by explicitly supervising the reasoning process through a Critical Evidence Graph (CEG). We curate a dataset of challenging clinical cases and algorithmically construct a CEG for each sample to represent a high-quality verifiable reasoning pathway. To guide the reasoning process, we introduce a Clinical Reasoning Procedure Reward, which evaluates Node Coverage, Structural Correctness, and Chain Completeness, thereby providing a holistic assessment of reasoning quality. Experimental results show that MedCEG achieves competitive performance across multiple medical benchmarks while generating more clinically sound reasoning chains, offering a promising step towards more reliable medical AI reasoning.

## 1 Introduction

Large Language Models (LLMs) with reasoning capabilities have demonstrated notable advances in various domains (Guo et al., 2025; Yang et al., 2025), with medical applications showing particular promise (Singhal et al., 2025; Tu et al., 2025). These models have achieved improved performance on complex medical tasks, including diagnostic reasoning (Liu et al., 2025b), treatment planning (Goh et al., 2025), and clinical question answering (Jin et al., 2021; Hager et al., 2024). Beyond performance gains, the explicit reasoning processes of models can serve as valuable support for human-AI collaboration in clinical scenarios (Liévin et al., 2024; Singhal et al., 2025), offering practitioners additional perspectives for decision-making and serving as effective educational aids.

To develop advanced reasoning capabilities in LLMs, researchers have explored various training approaches, with reinforcement learning (RL) emerging as a primary method (Ouyang et al., 2022). Early efforts often combined standard algorithms like Proximal Policy Optimization (PPO) (Schulman et al., 2017) with a Process Reward Model (PRM) (Lightman et al., 2023; Huang et al., 2025) to provide step-by-step supervision. While effective, the value function used in PPO typically imposes considerable memory and computational burden. In addition, collecting fine-grained supervised data for PRM is both time-consuming and labor-intensive (Rafailov et al., 2023). In contrast, more recent methods, such as Group Relative Policy Optimization (GRPO), have been developed to address these inefficiencies (Shao et al., 2024). By eliminating the auxiliary value model, GRPO significantly reduces computational overhead and memory footprint, making it more accessible and cost-effective to incentivize the reasoning capabilities of large-scale models (Guo et al., 2025).

However, a critical challenge arises when GRPO is paired with an outcome-oriented reward function, particularly in the medical domain where logical rigor is paramount (Singhal et al., 2025). When the optimization objective focuses primarily on maximizing the accuracy of final answers, models may learn to exploit misleading patterns in the data to take shortcuts. Such behavior results in conclusions that appear correct but are based on illogical or clinically invalid reasoning processes. This "right for the wrong reasons" phenomenon (McCoy et al., 2020) poses a significant safety risk.

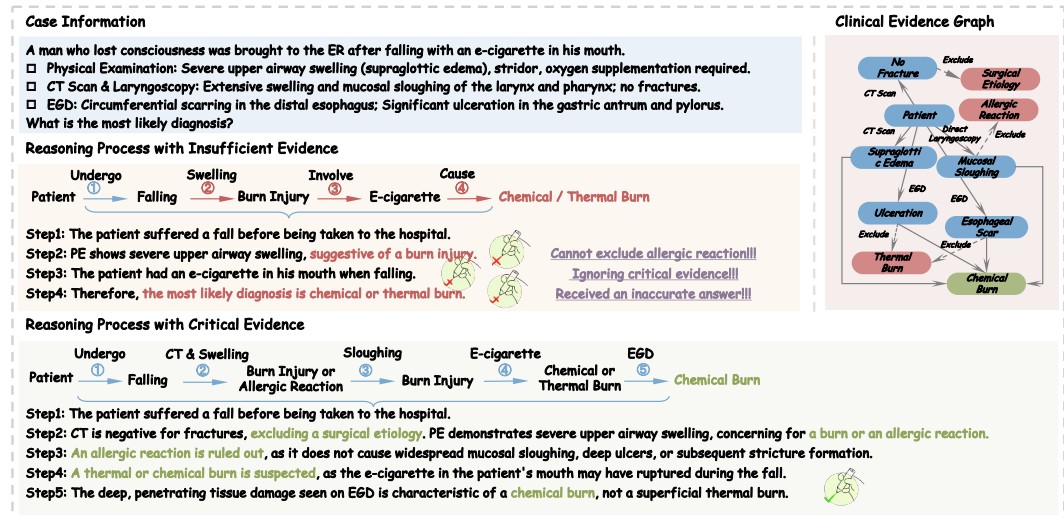

Figure 1: This diagram contrasts two types of reasoning process. The top process shows a flawed approach leading to a hasty conclusion. The bottom process, guided by the Clinical Evidence Graph (right), follows a systematic pathway to a more precise and reliable diagnosis.

As illustrated in Figure 1, this manifests as a defective diagnostic process where a model might leap to a hasty conclusion such as "chemical burn" or "thermal burn", while completely bypassing the crucial evaluation of key pathological findings. Although this shortcut in reasoning may happen to yield the correct result by coincidence, it contravenes the fundamental principles of evidence-based medicine and can adversely affect subsequent treatment decisions (Kim et al., 2025).

Building on these premises, we propose a graph-based alignment framework that both eliminates the need for costly PRM-training and bypasses the drawbacks of conventional outcome-oriented reward functions. By converting clinical narratives into traceable evidence graphs, we construct a structured and explicit representation of the reasoning process, serving as a direct, non-learned reward signal. Our approach begins by algorithmically constructing instance-specific *Evidence Graphs* (EGs), a process that externalizes the implicit, narrative logic of the text into an explicit and structured format. Based on EGs, we then extract a *Critical Evidence Graphs* (CEG) for each graph. This pathway captures essential clinical entities and their causal relationships, representing the minimal, logically necessary backbone of the argument. The EG and its refined CEG serve distinct purposes in our training pipeline. First, the EG is linearized back into a coherent textual sequence for Cold-Start (Guo et al., 2025), compelling the model to generate explanations that explicitly articulate the graph's formal, step-by-step logical trajectory. Second, the CEG provides a direct reward signal for a subsequent RL phase. This ensures reinforcement targets the most critical inferential steps, training models to prioritize verifiable causal pathways essential for sound clinical reasoning. Experiments demonstrate that our method significantly enhances the logical coherence of reasoning processes while improving overall answering accuracy, achieving SOTA performance with 58.59 on in-distribution tasks and 64.09 on out-of-distribution tasks. In summary, our contributions are:

- We construct and publicly release a dataset of Critical Evidence Graphs, providing a structured and explicit resource to guide clinical soundness. Our dataset contains 10K clinical cases with corresponding CEGs that explicitly model medical entities, their relationships, and causal pathways, facilitating the development of more reliable medical AI systems.

- We design a CEG-based reward mechanism that provides direct supervision for the entire reasoning process. This method compels the model to generate explanations that are not only accurate but also follow clinically valid and verifiable pathways.

- We demonstrate through extensive experiments that our method achieves state-of-the-art performance on multiple challenging medical question-answering benchmarks. Beyond performance gains, our approach produces more clinically sound reasoning chains, showing significant improvements in reasoning quality compared to existing methods.

## 2 RELATED WORK

### 2.1 REINFORCEMENT LEARNING INCENTIVES FOR LLM REASONING

RL has emerged as a key paradigm for eliciting systematic reasoning capabilities in LLMs by providing structured feedback on multi-step problem-solving processes (Yuan et al., 2024; Plaat et al., 2024). Early RLHF approaches (Ouyang et al., 2022) reinforced correct final answers but suffered from credit assignment challenges in multi-step scenarios, where sparse rewards make it difficult to identify which intermediate steps contribute to successful outcomes. PRMs address this limitation by providing dense feedback on each reasoning step, enabling models to learn step-by-step logical decomposition and error correction strategies (Lightman et al., 2023; Huang et al., 2025). Recent approaches have sought to eliminate complex external supervision while maintaining reasoning quality. A prominent example is Direct Preference Optimization (DPO), which simplifies the pipeline by directly optimizing on preference data without a separate reward model (Rafailov et al., 2023). Besides, GRPO train models to generate and compare multiple reasoning paths, learning to distinguish high-quality logical chains through relative ranking (Shao et al., 2024; Guo et al., 2025).

### 2.2 STRATEGIES FOR TRUSTWORTHY MEDICAL REASONING

Establishing reliable reasoning in medical LLMs demands both transparency and robustness in their decision-making processes. Foundational approaches such as Chain-of-Thought (CoT) (Wei et al., 2022) prompting address this challenge by leveraging curated reasoning pathways derived from expert medical resources. For instance, MedReason (Wu et al., 2025a) leverages the authoritative knowledge graph PrimeKG (Chandak et al., 2023) to create reasoning chains supported by comprehensive evidence, making the model's logic auditable. However, simply showing a model static examples is not enough. Advanced techniques like PRMs provide fine-grained, step-by-step evaluation and feedback. Rather than merely presenting correct reasoning pathways, these models assess each individual logical step, directly incentivizing the construction of clinically sound and coherent reasoning sequences (Huang et al., 2025). Models like Huatuo-o1 (Chen et al., 2024) and MedS$^3$ (Jiang et al., 2025) showcase this advanced paradigm, employing sophisticated reward mechanisms to refine the entire reasoning process. This represents a shift from simply demonstrating correct reasoning patterns to actively training models to develop reliable reasoning capabilities.

## 3 DATA PREPARATION

In this paper, we curate a clinical rationale corpus that combines challenging clinical questions with structured reasoning representations for developing rigorous reasoning capabilities. Our pipeline involves: filtering difficult cases from existing medical question-answer datasets, enriching cases with high-quality rationales, transforming rationales into structured evidence graphs, and extracting the clinical evidence graphs that connects the initial evidence to the final conclusions. Detailed construction procedures and validation results are provided in the Appendix C.

**Clinical Rationale Corpus Curation** Our process begins with a raw dataset $\mathcal{D}_{\text{raw}}$ aggregated from MedQA (Jin et al., 2021), MedCase (Wu et al., 2025b), and JAMA Challenge sources. To isolate the challenging cases, we employ an ensemble-based filtering method (Ankner et al., 2024; Mizrahi et al., 2025). Specifically, for each question, we use *Llama-3.1-8B-Instruct* (Dubey et al., 2024) to make 16 independent generation attempts. A question-answer pair is retained only if the model generates the correct answer in fewer than half of the attempts (i.e., a consensus score $S < 0.5$), forming the hard dataset. For each question in the hard dataset, we then use *Gemini-2.5-Pro* (Comanici et al., 2025) to generate a high-quality question-rational-answer triplet $(q, r, a)$. We employ an iterative sampling strategy with up to 4 attempts, accepting a triplet only if its answer is correct. This process yields our curated corpus, $\mathcal{D}_{\text{curated}} = \{(q_i, r_i, a_i)_{i=1}^N\}$.

**Evidence Graph Construction** Next, we transform each textual rationale $r$ from $\mathcal{D}_{\text{curated}}$ into a structured Evidence Graph $G$. We employ an ensemble of three diverse models (i.e., *GPT-OSS-120B* (Agarwal et al., 2025), *Qwen3-235B* (Yang et al., 2025), and *DeepSeek-V3* (Liu et al., 2024)) to extract candidate reasoning triplets from each rationale. Specifically, we design a specialized prompt to guide these models in extracting semantic relationships as subject-predicate-object triplets

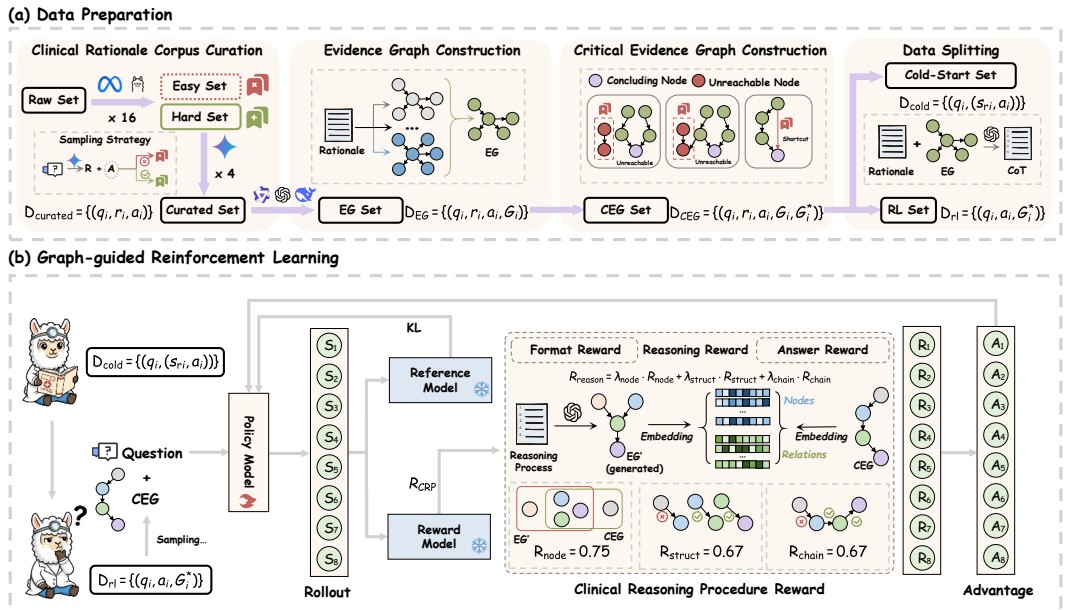

Figure 2: Overview of the MedCEG pipeline. (a) Dataset preparation: Illustrates the process of curating a clinical rationale corpus and subsequently constructing the Evidence Graph and Critical Evidence Graph. (b) Graph-guided RL: A policy model generates reasoning paths and is optimized using a Clinical Reasoning Procedure Reward, which is guided by the Critical Evidence Graph.

$t = (s, p, o)$ that capture logical connections between reasoning concepts. A triplet is accepted and used to build the graph only if it is extracted by at least two of the three models. This entire process yields our dataset with EGs, $\mathcal{D}_{EG} = \{(q_i, r_i, a_i, G_i)_{i=1}^N\}$, where $G_i = \{(s, p, o)\}$ is the set of accepted reasoning triplets for $r_i$. In essence, this step robustly converts free-form text into a structured representation of its underlying logic.

---

**Two Forms of Our Graph Data Structure: $G$ and $\mathcal{G}$**

Our graph data can be modeled into two interchangeable structures: a set of triplets or a directed graph. The triplets $G = \{(s, p, o)\}$ is converted into directed graph $\mathcal{G} = (\mathcal{V}, \mathcal{E})$ by:
- **Vertices** $\mathcal{V}$: The set of all unique entities ($s$ or $o$).
- **Edges** $\mathcal{E}$: Each triplet defines a directed edge $(s, o)$, with $p$ as its semantic label.

---

**Critical Evidence Graph Extraction**   To highlight key reasoning components, we introduce the Critical Evidence Graph $G^*$ as a refined subgraph that captures the essential reasoning pathway within the broader EG, containing only the most granular and logically necessary components for reaching the correct conclusion. Given a $G$, the construction of $G^*$ begins by using *GPT-OSS-120B* to identify a conclusion node with maximal semantic similarity to the ground-truth answer, from which we perform a backward traversal in EG $\mathcal{G}$ to extract the entire causally connected subgraph. This subgraph is then refined through transitive reduction, which prunes shortcuts (e.g., $A \rightarrow C$) in favor of the more granular, multi-hop paths that render each inferential step explicit (e.g., $A \rightarrow B \rightarrow C$). The final output is $G^*$, faithfully representing the parsimonious yet complete reasoning pathway from the initial evidence to the final conclusion, achieving the balance between comprehensiveness and conciseness. We then built the dataset $\mathcal{D}_{CEG} = \{(q_i, r_i, a_i, G_i, G_i^*)_{i=1}^N\}$.

**Data Splitting**   To facilitate the two-stage progressive learning paradigm (Guo et al., 2025), our final dataset is first partitioned into a Cold-Start subset $\mathcal{D}_{cold}$ and a reinforcement learning subset $\mathcal{D}_{rl}$ via stratified sampling based on CEG complexity. For the preparation of the Cold-Start data, we must bridge the gap between the structured reasoning graphs and the sequential format required by language models. To achieve this, we employ *GPT-OSS-120B* to systematically transform each

evidence graph $G$ into a coherent natural language reasoning sequence $s$ that preserves logical dependencies and clinical precision. The Cold-Start data follows the format $(q_i, (s_i, a_i))$, while $\mathcal{D}_{\text{rl}}$ is structured as $(q_i, G_i^*, a_i)$, directly providing the question, the final answer, and the core reasoning subgraph to guide the reinforcement process.

# 4 GRAPH-GUIDED REINFORCEMENT LEARNING

To ensure the logical soundness of our model's reasoning, we employ a two-stage progressive learning paradigm. The first stage establishes foundational capabilities through a Cold-Start stage on $\mathcal{D}_{\text{cold}}$, a standard methodology detailed in Appendix D. Building on this, we introduce a Critical Evidence Graph-guided Reinforcement Learning approach. Instead of solely relying on outcome-based rewards, we introduce a novel process-oriented reward function that leverages CEG $G^*$ to evaluate reasoning quality across multiple complementary dimensions.

## 4.1 CLINICAL REASONING PROCEDURE REWARD

To ensure that the model's generated reasoning is clinically comprehensive, factually accurate, and logically coherent, we designed a composite reward signal, which we term the **Clinical Reasoning Procedure** (CRP) reward. This signal integrates three distinct components that evaluate the quality of the reasoning process, the accuracy of the final answer, and the adherence to a specified format. The cornerstone of our reward model is the Process Reward ($R_{\text{reason}}$), which holistically assesses the intrinsic quality of the generated reasoning chain. It is calculated from three complementary metrics: Node Coverage, Structural Correctness, and Reasoning Chain Completeness. These metrics systematically evaluate reasoning quality by ensuring conceptual comprehensiveness, factual accuracy of relationships, and overall logical coherence. Notably, to evaluate the generated reasoning, we parse the model's textual thinking process and extract triplets to form the generation EG $\tilde{G}$, using GPT-OSS-120B by the methodology described in Section 3.

**Node Coverage ($R_{\text{node}}$)** The Node Coverage score evaluates whether the model's reasoning incorporates all essential clinical concepts. This metric quantifies the semantic coverage of the ground-truth entities ($\mathcal{V}^*$) from the CEG $\mathcal{G}^*$ by the set of generated entities ($\tilde{\mathcal{V}}$):

$$R_{\text{node}} = \frac{1}{|\mathcal{V}^*|} \sum_{u \in \mathcal{V}^*} \max_{v \in \tilde{\mathcal{V}}}\{\text{sim}(u, v)\}, \tag{1}$$

where $\text{sim}(u, v)$ is defined as the cosine similarity between the vector representations of the clinical concepts $u$ and $v$. These dense vector embeddings are generated using the pre-trained *bge-large-en-v1.5* (Xiao et al., 2023). A high $R_{\text{node}}$ score indicates that the model's reasoning is built upon a comprehensive evidential foundation.

**Structural Correctness ($R_{\text{struct}}$)** The Structural Correctness score assesses the factual accuracy of the relationships established between clinical concepts. It computes the recall ratio of ground-truth triplets, where a triplet $t^* = (s, p, o) \in G^*$ is considered recalled if it exists in the generated graph $\tilde{G}$, captured by an indicator function $\mathbb{I}(\cdot)$ that returns 1 if the condition is true and 0 otherwise, i.e.,

$$R_{\text{struct}} = \frac{1}{|G^*|} \sum_{t^* \in G^*} \mathbb{I}(t^* \in \tilde{G}). \tag{2}$$

This metric penalizes outputs that link correct concepts in a factually incorrect manner, thereby ensuring the structural integrity of the reasoning process.

**Chain Completeness ($R_{\text{chain}}$)** This metric assesses logical coherence by measuring the proportion of ground-truth triplets that form a single, unbroken line of reasoning. We first identify the set of recalled triplets to construct an undirected graph $\mathcal{G}$ from these recalled triplets and find its largest connected component $C_{\text{max}}$. The reward is the fraction of all ground-truth triplets contained within:

$$R_{\text{chain}} = \frac{|C_{\text{max}} \cap G^*|}{|G^*|} \tag{3}$$

A high $R_{\text{chain}}$ score indicates that the model has successfully woven the facts into a logically consistent narrative from evidence to conclusion.

**Final Reward Calculation** To obtain a comprehensive assessment of reasoning quality, these three metrics are combined to form the final $R_{\text{reason}}$ signal, calculated as:

$$R_{\text{reason}} = \lambda_{\text{node}} \cdot R_{\text{node}} + \lambda_{\text{struct}} \cdot R_{\text{struct}} + \lambda_{\text{chain}} \cdot R_{\text{chain}}, \tag{4}$$

where $\lambda_{\text{node}}$, $\lambda_{\text{struct}}$ and $\lambda_{\text{chain}}$ are the weights for the node, structure, and chain rewards, respectively. Finally, this comprehensive reasoning score is integrated with simple binary rewards for final answer accuracy and adherence to specified output format, forming a complete reward signal that ensures the model is optimized for logical soundness, factual correctness, and stylistic consistency.

## 4.2 OPTIMIZATION

We use the composite CRP reward signal to fine-tune the model using GRPO algorithm. GRPO iteratively refines the language model's policy, $\pi_\theta$, to maximize the expected reward while maintaining stability by penalizing large deviations from a reference policy, $\pi_{\text{ref}}$. The core of the algorithm is its objective function, which adapts the clipped surrogate objective from PPO. Let the probability ratio for a token be $r_t(\theta) = \frac{\pi_\theta(o_{i,t}|q,o_{i,<t})}{\pi_{\theta_{\text{old}}}(o_{i,t}|q,o_{i,<t})}$, where $\pi_{\theta_{\text{old}}}$ is the policy before the update. The GRPO objective $\mathcal{J}_{\text{GRPO}}(\theta)$ is then defined as:

$$\mathcal{J}_{\text{GRPO}}(\theta) = \mathbb{E}\left[ q \sim \mathcal{D}_{\text{rl}}, \{o_i\}_{i=1}^{G} \sim \pi_{\theta_{\text{old}}}(O \mid q) \right]$$

$$\frac{1}{G} \sum_{i=1}^{G} \frac{1}{|o_i|} \sum_{t=1}^{|o_i|} \left\{ \min\left[ r_t(\theta)\hat{A}_{i,t}, \text{clip}\left(r_t(\theta), 1-\varepsilon, 1+\varepsilon\right)\hat{A}_{i,t} \right] - \beta \mathbb{D}_{\text{KL}}\left[\pi_\theta || \pi_{\text{ref}}\right] \right\}. \tag{5}$$

where $r_t(\theta)$ is the probability ratio between the current policy $\pi_\theta$ and the old policy $\pi_{\theta_{\text{old}}}$, and $\hat{A}_{i,t}$ is the estimated advantage. The clip function constrains $r_t(\theta)$ within the range $[1-\varepsilon, 1+\varepsilon]$. $\mathbb{D}_{\text{KL}}\left[\pi_\theta || \pi_{\text{ref}}\right]$ represents the KL divergence between the current policy and a reference policy $\pi_{\text{ref}}$, with $\beta$ as its penalty coefficient. By optimizing this combined objective, GRPO effectively aligns the model's reasoning with ground-truth clinical pathways while ensuring its outputs remain factually accurate and structurally sound.

## 5 EXPERIMENTS

### 5.1 EXPERIMENTAL SETUP

**Benchmarks** For a rigorous and multifaceted assessment of our model's medical capabilities, we performed evaluations on a suite of established medical QA benchmarks. These benchmarks cover both multiple-choice and open-ended question formats. Our evaluation is structured into two primary domains: in-domain benchmarks, consisting of (i) **MedQA** (Jin et al., 2021), (ii) **MedBullets-5op** (Chen et al., 2025), and (iii) **MedCase** (Wu et al., 2025b); and out-of-domain benchmarks, including (i) **MMLU-H** (Hendrycks et al., 2020), (ii) **MMLU-Pro-H** (Wang et al., 2024), and (iii) **DiagArena** (Zhu et al., 2025). Among these, MedCase is in an open-ended question format, while the rest are multiple-choice questions. Further details on these are available in Appendix E.1.

**Baselines** We benchmark our proposed MedCEG against a range of existing models. The baselines include the foundational model, *Gemma3-4B-it* (Team et al., 2025), *Llama3.1-8B-Instruct* (Dubey et al., 2024) and *Qwen2.5-7B-Instruct* (Qwen et al., 2025), as well as models that have been continually trained: *MedGemma-4B-it* (Sellergren et al., 2025), *OpenBioLLM-8B* (Ankit Pal, 2024), *UltraMedical3.0-8B* (Zhang et al., 2024), and *UltraMedical3.1-8B* (Zhang et al., 2024). Furthermore, we compare several models specifically enhanced for medical reasoning: *MedS³-8B* (Jiang et al., 2025) and *Huatuo-o1-8B* (Chen et al., 2024), trained with a PRM; *MedReason-8B* (Wu et al., 2025a), fine-tuned on Huatuo-o1-8B using a human-designed chain-of-thought approach; and *AlphaMed-8B* (Liu et al., 2025a), utilized GRPO with outcome-supervised reward.

**Evaluation** Our evaluation assesses both the final outcome and the reasoning process. For outcomes, we measure Accuracy on multiple-choice questions (MCQs) and use *GPT-OSS-120B* to judge the final answers on open-ended tasks. For the reasoning process, we employ a committee of three models—*DeepSeek-R1*, *Qwen3-235B*, and *GPT-5-High*—to ensure a robust assessment. This

Table 1: A comprehensive performance comparison of various medical language models across a diverse suite of benchmarks. **Bold** denotes the best result, and an underline indicates the second-best. "**–**" signifies that the model was unable to generate a valid or evaluable response for the task.

| Model | In-Distribution | | | | Out-of-Distribution | | | |
|---|---|---|---|---|---|---|---|---|
| | **MedQA** | **MedBullets-5op** | **MedCase** | **Average** | **MMLU-H** | **MMLU-Pro-H** | **DiagArena** | **Average** |
| *General Models* | | | | | | | | |
| Gemma3-4B-it | 38.88 | 37.01 | 11.82 | 29.24 | 44.63 | 24.94 | 21.53 | 30.37 |
| Qwen2.5-7B-Instruct | 54.28 | 37.99 | 15.16 | 35.81 | 69.24 | 49.39 | 21.96 | 46.86 |
| Llama-3.1-8B-Instruct | 61.59 | 44.48 | 16.83 | 40.97 | 73.00 | 51.71 | 35.63 | 53.45 |
| *Medical Instruct Models* | | | | | | | | |
| MedGemma-4B-it | 63.16 | 45.45 | 14.05 | 40.89 | 66.30 | 43.28 | 40.44 | 50.01 |
| OpenBioLLM-8B | 43.75 | 32.79 | 12.37 | 29.63 | 54.64 | 28.61 | 39.34 | 40.86 |
| UltraMedical3.0-8B | 61.57 | 41.56 | 14.94 | 40.69 | 66.67 | 53.30 | 34.10 | 53.02 |
| UltraMedical3.1-8B | 73.21 | 54.55 | 15.38 | 47.71 | 77.04 | 57.82 | 38.69 | 57.85 |
| *Medical Reasoning Models* | | | | | | | | |
| MedS$^3$-8B | 71.88 | 49.68 | 18.84 | 46.79 | 79.50 | 57.21 | 36.50 | 57.73 |
| MedReason-8B | 71.80 | 55.50 | 19.06 | 48.79 | 75.76 | 57.09 | 36.83 | 56.56 |
| AlphaMed-8B | **75.41** | 61.69 | – | 45.70 | 79.87 | **65.28** | 37.16 | 60.77 |
| Huatuo-o1-8B | 72.60 | 53.90 | 18.39 | 48.30 | 79.34 | 58.70 | 49.18 | 62.41 |
| *Our Models* | | | | | | | | |
| MedCEG (only SFT) | 73.06 | 58.77 | 28.09 | 56.38 | 79.25 | 59.54 | 48.63 | 62.47 |
| MedCEG (Cold Start) | 71.96 | 59.09 | 28.65 | 55.59 | 80.44 | 57.82 | 47.54 | 61.93 |
| MedCEG | **75.41** | **63.64** | **31.55** | **58.59** | **81.18** | 62.22 | **50.16** | **64.09** |

committee scores the reasoning quality against five criteria: (i) **Logical Coherence**, (ii) **Factual Accuracy**, (iii) **Evidence Faithfulness**, (iv) **Interpretability & Clarity**, and (v) **Information Utilization**. Furthermore, a human expert consistency analysis was performed to ensure the reliability of this automated process evaluation. Detailed rubrics and prompts are available in Appendix E.3.

**Implementation Details**  All experiments are conducted on 8 NVIDIA H100 GPUs using the LLaMA-Factory (Zheng et al., 2024) and VERL (Sheng et al., 2024) framework. In the Cold Start Stage, we perform full-parameter fine-tuning on *Llama-3.1-8B-Instruct* with a learning rate of $1 \times 10^{-6}$, optimized using DeepSpeed ZeRO-2. The RL Stage is optimized using GRPO with a learning rate of $5 \times 10^{-7}$. A KL-divergence penalty with a coefficient $\beta = 0.001$ is used to regularize the policy against the SFT model. Both stages adopt a cosine learning rate decay schedule.

## 5.2 BENCHMARKING RESULTS

To rigorously evaluate the capabilities of our proposed MedCEG model, we conducted comprehensive benchmarking against a curated set of LLMs. As detailed in Table 1, these evaluations encompass both In-Distribution (ID) and Out-of-Distribution (OOD) scenarios, providing a holistic assessment of the model's specialized domain expertise and generalization capacity. The results establish MedCEG as the new state-of-the-art model, achieving superior performance with average scores of $58.59\%$ on ID tasks and $64.09\%$ on OOD tasks.

Specifically, MedCEG demonstrates remarkable improvements over mainstream general-purpose models, achieving substantial performance gains compared to *Llama-3.1-8B-Instruct*, underscoring the critical importance of domain-specific optimization in medical applications where nuanced understanding of clinical contexts and medical terminology is paramount. When benchmarked against specialized medical instruct models, MedCEG maintains its competitive edge with notable consistency. This performance differential suggests that while standard instruction tuning on medical dialogue datasets enhances question-answering capabilities, our training paradigm cultivates the deeper, more sophisticated reasoning abilities essential for complex clinical decision-making scenarios. The superiority of MedCEG becomes even more pronounced when compared against models explicitly designed for complex medical reasoning, such as *AlphaMed-8B* and *Huatuo-o1-8B*. Notably, Med-CEG outperforms the competitive *Huatuo-o1-8B* by $10.29$ on ID tasks and $1.68$ on OOD tasks, demonstrating consistent excellence across diverse evaluation scenarios.

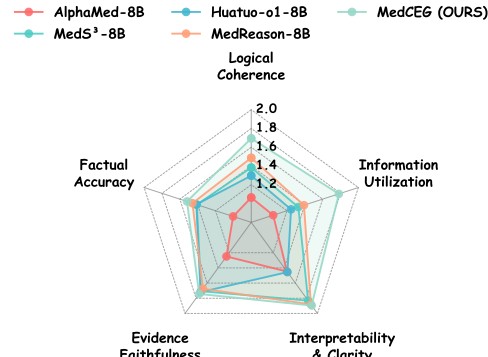 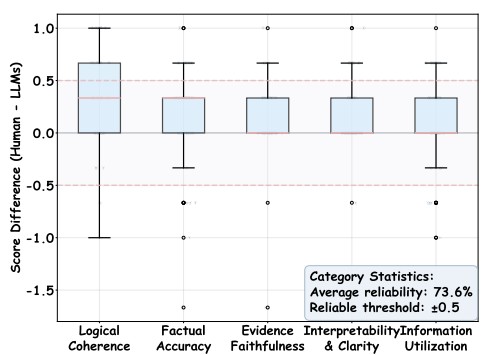

Figure 3: Multi-dimensional evaluation of the reasoning process for different models. This chart compares the quality of the reasoning process for MedCEG against four baseline models across five criteria.

Figure 4: Box plot analysis of scoring consistency between human and LLM evaluations. The y-axis represents the score difference, where positive values indicate higher scores from humans and negative values indicate lower scores.

The efficacy of our proposed Clinical Reasoning Process (CRP) reward paradigm is comprehensively validated through MedCEG's superior performance across various benchmarks. In MedQA which focused on medical knowledge, MedCEG achieves $75.41\%$, matching top-performing *AlphaMed-8B* while outperforming other medical reasoning models like Huatuo-o1-8B ($72.60\%$). The advantage becomes more pronounced on MedBullets-5op ($63.64\%$ vs. *AlphaMed-8B*'s $61.69\%$) and most remarkable on the challenging open-ended MedCase benchmark, where MedCEG achieves $31.55\%$, substantially exceeding all competing models while *AlphaMed-8B* failed entirely to generate valid outputs. This progressive performance enhancement from standard medical QA to complex clinical reasoning and comprehensive case analysis validates our core hypothesis: supervising and optimizing the reasoning process itself becomes increasingly crucial as task complexity grows, particularly for sophisticated clinical thinking requiring complete and coherent reasoning chains.

### 5.3 REASONING PROCESS ASSESSMENT

For a fine-grained assessment of the models' reasoning quality, a cohort of 2000 correctly adjudicated instances was randomly sampled from each model under review. The latent reasoning trajectories inherent in these responses were subsequently subjected to systematic scoring by a consortium of evaluator models, benchmarked against five pivotal dimensions.

Our MedCEG model demonstrates clear superiority across all evaluated dimensions, as shown in Figure 3. Achieving an aggregate score of $8.64$, MedCEG outperforms the strongest baseline, *MedReason-8B* ($7.89$), representing a $9.5\%$ improvement. While models such as *MedReason-8B*, which was fine-tuned on meticulously curated datasets, and those employing PRM for reinforcement learning, namely *Huatuo-o1-8B* and *MedS³-8B*, deliver competitive performance, they do not attain the holistic excellence demonstrated by MedCEG. A particularly salient observation pertains to the performance of *AlphaMed-8B*: notwithstanding its commendable accuracy on multiple-choice benchmarks, the fidelity of its reasoning process is consistently ranked as subordinate, scoring as low as 0.98 in Logical Coherence. This dichotomy starkly illuminates the inherent limitations of supervision paradigms predicated exclusively on final-outcome optimization. In aggregate, these findings affirm the superior capacity of MedCEG to generate reasoning chains characterized by enhanced quality, trustworthiness, and interpretability relative to its contemporaries. For qualitative elucidation, illustrative case studies are meticulously detailed in Appendix G. By compelling models to follow logically rigorous pathways, our framework represents a meaningful step toward developing safer and more reliable AI systems in healthcare.

Besides, to validate the consistency between the automated scoring and human expert evaluation, we randomly sampled 100 instances from each model's generated data (for a total of $500$ instances). These were then scored by human experts, and a consistency analysis was performed against the average scores from the evaluator models. We performed an in-depth consistency analysis by calculating the score difference for each evaluation. To quantify the level of agreement, we established

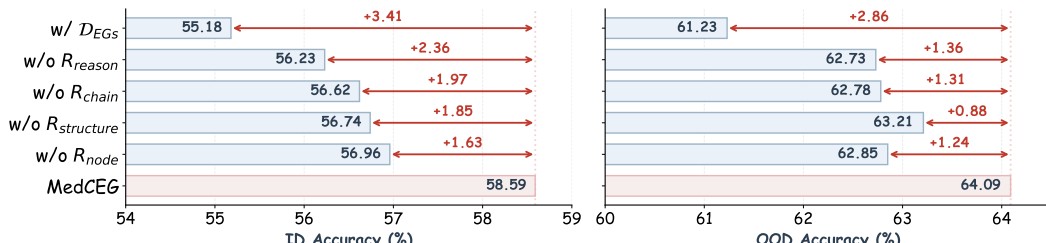

Figure 5: Ablation analysis of different components' impact. We compare our full model against several variants: w/ $\mathcal{D}_{\text{EGs}}$ uses EG data for RL instead of CEG data, w/o $R_{\text{reason}}$ removes the entire reasoning reward, w/o $R_{\text{chain}}$, w/o $R_{\text{structure}}$, and w/o $R_{\text{node}}$ ablate the reward modules for chain completeness, structural correctness, and node coverage, respectively.

a Reliable Threshold of $\pm 0.5$, defining any score difference within this range as consistent. The results, as illustrated in Figure 4, show a significant positive correlation between the automated and human expert scores. Specifically, the average reliability across all evaluations reached 73.6%, meaning nearly two-thirds of the automated scores fell within the acceptable range of the human scores. Collectively, these findings confirm a high and reliable degree of consistency between our automated tool and human expert judgment, validating its effectiveness as a scalable evaluation tool.

### 5.4 ABLATION STUDY

**Ablation Study on the Training Pipeline**   Table 1 delineates the contribution of each training phase. First, the SFT-only version of our model, MedCEG (only SFT), establishes an exceptionally strong baseline. Quantitatively, it achieves an average score of $56.38$ on ID tasks and $62.47$ on OOD tasks, surpassing nearly all other leading models. Second, the subsequent reasoning alignment stage consistently elevates this powerful baseline. Within our two-stage training framework, the model's performance progressively improves, ultimately surpassing the SFT-only baseline. This outcome unequivocally validates our multi-stage training architecture and highlights the critical role of the second stage in consolidating and refining the model's advanced reasoning capabilities.

**Ablation Study on the Reasoning Reward**   We conduct a detailed ablation study to validate the design of our composite reward function in the MedCEG model. This analysis quantifies the contribution of each individual component to the model's overall performance. As illustrated in Figure 5, the model's overall performance declines when any of the individual reward components, namely $R_{\text{node}}$, $R_{\text{structure}}$, and $R_{\text{chain}}$, are removed from the reasoning reward. Notably, the removal of $R_{\text{chain}}$ results in the most significant performance degradation among the three (OOD Accuracy $\Delta = 1.31$, ID Accuracy $\Delta = 1.97$), highlighting the critical role of maintaining logical coherence and step-by-step correctness throughout the reasoning process. The variant w/o $R_{\text{think}}$, which relies solely on outcome-based supervision without reasoning-based rewards, exhibits only a marginal improvement over the initial training stage, underscoring the insufficiency of coarse-grained, outcome-only supervision for refining complex reasoning pathways. Furthermore, replacing our CEG-based reward with the more structurally complex EG-based reward (denoted as w/ $\mathcal{D}_{\text{EGs}}$) leads to a catastrophic collapse in performance, a decline potentially attributable to an overabundance of supervisory signals within the EG framework that can stifle the model's exploratory reasoning capabilities.

## 6 CONCLUSION

In this work, we addressed the critical challenge of flawed reasoning pathways in medical Large Language Models, which often learn shortcuts due to outcome-focused supervision. To tackle this, we introduced MedCEG, a novel graph-based framework designed to explicitly supervise the entire reasoning process. Our approach algorithmically constructs Evidence Graphs from clinical narratives, which are linearized to guide an initial Cold-Start training phase. Subsequently, we extract Critical Evidence Graphs to provide a direct, non-learned reward signal for a reinforcement learning stage. our method achieves state-of-the-art performance on multiple medical benchmarks while producing more clinically sound and verifiable reasoning chains.

## REPRODUCIBILITY STATEMENT

To facilitate the reproduction of our results, we have made our source code, data samples, and model weights anonymously available at `https://anonymous.4open.science/r/MedCEG-CDBD`. We provide a comprehensive overview of our data construction methodology, detailing all data sources, the full processing pipeline, and the specific prompts used, in Appendix C and D. Furthermore, the experimental setup, including the training details and hyperparameter configurations for both stages of our method, is presented in Appendix F. The Supplementary Materials contain training data examples, key training code, reasoning process evaluation scripts, benchmark performance results of MedCEG, inference demonstrations, and other relevant materials.

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

## A  LLM USAGE STATEMENT

During the preparation of this manuscript, we utilized a large language model (LLM) as a general-purpose tool to assist with grammar checking and language refinement. In our experimental process, an LLM was employed for two specific tasks: (i) in data preparation, as described in Section 3, and (ii) to evaluate reasoning processes as part of our evaluation protocol, detailed in Section 5. **The authors have carefully reviewed and edited all LLM-generated content and take full responsibility for the final version of this paper.**

## B  SYMBOL DESCRIPTION

Table 2: Notation and Symbol Definitions

| Symbol | Description |
| --- | --- |
| **Dataset Variables** | |
| $\mathcal{D}_{\text{raw}}$ | Raw dataset |
| $\mathcal{D}_{\text{curated}}$ | Curated corpus |
| $\mathcal{D}_{\text{EG}}$ | Dataset with Evidence Graphs |
| $\mathcal{D}_{\text{CEG}}$ | Dataset with Critical Evidence Graphs |
| $\mathcal{D}_{\text{cold}}$ | 80% subset of $\mathcal{D}_{\text{CEG}}$ used for supervised fine-tuning |
| $\mathcal{D}_{\text{rl}}$ | 20% subset of $\mathcal{D}_{\text{CEG}}$ used for reinforcement learning |
| **Graph Representations** | |
| $G$ | Evidence Graph |
| $G^*$ | Critical Evidence Graph |
| $\mathcal{G} = (\mathcal{V}, \mathcal{E})$ | Directed graph with vertices $\mathcal{V}$ and edges $\mathcal{E}$ |
| $t = (s, p, o)$ | Triplet with subject, predicate, and object |
| $\tilde{\cdot}$ | Generated content |
| $C_{\text{max}}$ | Largest connected component in recalled triplets graph |
| **Loss and Objective Functions** | |
| $\mathcal{L}_{\text{SFT}}(\theta)$ | Supervised fine-tuning loss function |
| $J_{\text{GRPO}}(\theta)$ | GRPO objective function |
| **Reward Components** | |
| $R_{\text{CRP}}$ | Clinical Reasoning Procedure composite reward |
| $R_{\text{reason}}$ | Reasoning Process Reward |
| $R_{\text{node}}$ | Node Coverage reward |
| $R_{\text{struct}}$ | Structural Correctness reward |
| $R_{\text{chain}}$ | Reasoning Chain Completeness reward |
| $R_{\text{answer}}$ | Binary reward for final answer accuracy |
| $R_{\text{format}}$ | Binary reward for adherence to output format |
| **Policy Optimization Variables** | |
| $\pi_\theta$ | Policy model with parameters $\theta$ |
| $\pi_{\text{ref}}$ | Reference policy model |
| $\pi_{\theta_{\text{old}}}$ | Policy model before parameter update |
| $r_t(\theta)$ | Probability ratio for token $t$ between current and old policy |
| $\hat{A}_{i,t}$ | Estimated advantage for trajectory $i$ at time step $t$ |

*Continued on next page*

| Symbol | Description |
|---|---|
| **Table 2 – continued from previous page** | |
| $D_{\text{KL}}[\pi_\theta || \pi_{\text{ref}}]$ | KL divergence between current and reference policies |
| **Parameters and Functions** | |
| $\text{sim}(u, v)$ | Cosine similarity between clinical concept embeddings |
| $\mathbb{I}(\cdot)$ | Indicator function returning 1 if condition is true, 0 otherwise |
| $S_{\text{weighted}}$ | Weighted score combining node, structure, and chain rewards |
| $\lambda_{\text{node}}, \lambda_{\text{struct}}, \lambda_{\text{chain}}$ | Weights for node, structure, and chain rewards respectively |
| $\epsilon$ | Clipping parameter for probability ratio |
| $\beta$ | KL divergence penalty coefficient |

## C  DETAILS OF DATASET PREPARATION

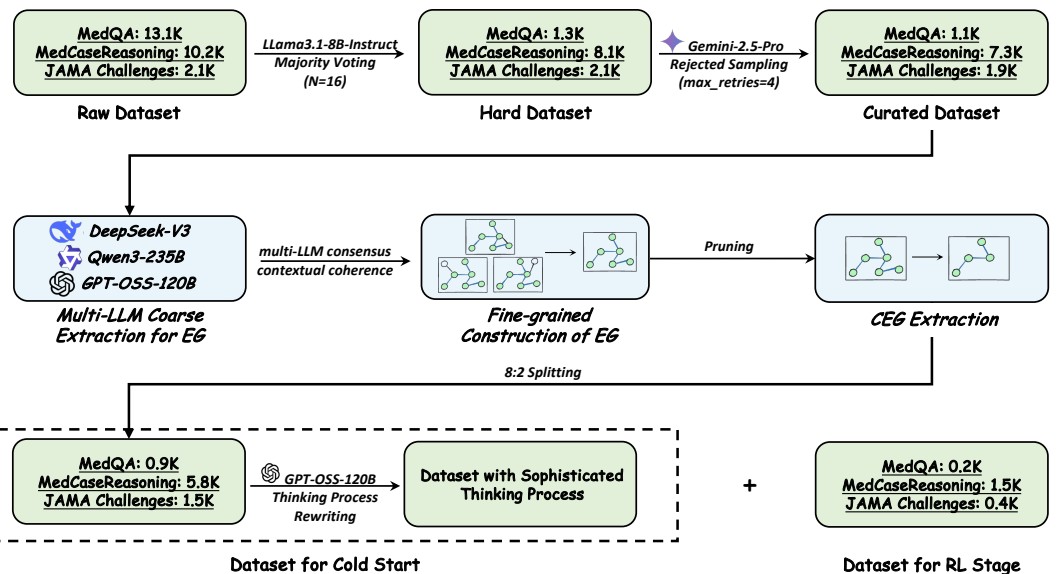

Figure 6: Detailed Construction Pipeline of Training Data

In this section, we provide a detailed overview of the data construction methodology, the prompts employed, and our validation experiments. The complete workflow is depicted in Figure 6.

### C.1  DETAILS OF CLINICAL RATIONALE CORPUS CURATION

First, to construct the Hard Dataset, we first implement a filtering protocol on the Raw Dataset $\mathcal{D}_{\text{raw}} = \{(q_i, a_i)\}$ to isolate the most challenging clinical scenarios. This process involves running inference on each instance using *Llama3.1-8B-Instruct* for $N = 16$ independent iterations with the prompt detailed below. Any instance that is answered correctly in more than half of these trials is considered non-trivial and is subsequently excluded. This selection methodology ensures that the Hard Dataset is populated exclusively with complex cases that demand robust reasoning.

---

**Prompt to Construct the Hard Dataset**

Question: {{Question}}

Please reasoning step by step, and put your final answer in \boxed{}.

---

Subsequently, we employ *Gemini-2.5-Pro* to perform rejection sampling on the dataset $\mathcal{D}_{\text{hard}}$. For each instance, we prompt the model to generate both a predicted answer and its analytical reasoning. This process is repeated up to a maximum of 4 times, and we retain the first correct response along with its corresponding analysis as dataset $\mathcal{D}_{\text{curated}} = \{(q_i, r_i, a_i)\}$. The specific prompt utilized for this procedure is as follows:

---

**Prompt to Construct the Curated Dataset**

Provide a comprehensive and well-reasoned answer to the following question. Your analysis must be thorough, leading logically to a definitive conclusion.

Question: {{Question}}

Instructions:

1. Deconstruct the Question: Begin by breaking down the question into its core components and defining any key terms to establish a clear foundation for your analysis.
2. Develop a Comprehensive rationale:
   • Present a detailed, step-by-step analysis that methodically addresses all parts of the question. Your reasoning must be transparent and easy to follow.
   • For Case-Specific Questions: Dissect the case by identifying the critical facts, underlying principles, and relevant context. Analyze how these elements interact to logically lead to the outcome.
   • Ensure your entire rationale forms a coherent and persuasive argument that directly supports your final conclusion.
3. Formulate the final_answer:
   • Start with a concise summary of the main points from your rationale.
   • Conclude with a direct and unambiguous answer to the original question.
   • The final, conclusive answer must be enclosed within the \boxed{...}.
Output Format:

Your entire response must be a single JSON object with the keys "final_answer" and "rationale", strictly adhering to the following structure:
{"final_answer": ..., "rationale": ...}

---

**Experimental Validation of Rationale Data** To validate the reasonableness and comprehensiveness of the rationales generated by rejection sampling approach, we employed a methodology akin to that described in Wu et al. (2025b). This involved extracting key reasoning evidence from the source papers corresponding to each case and subsequently computing the recall rate by identifying statements within the generated rationales that align with this evidence. We randomly sampled 1,000 instances from our dataset that possessed a source paper and utilized *GPT-OSS-120B* to perform the recall calculation. **The average recall rate achieved was 0.8213, which serves as a robust indicator of the rigor and accuracy of the generated reasoning processes.**

## C.2 Details of Evidence Graphs Construction

To construct a comprehensive reasoning graph for each process, we employ the meticulously designed prompt detailed below for extraction. This procedure involves retaining triplets that exhibit high consistency across multiple LLMs, ultimately yielding the dataset $\mathcal{D}_{\text{EG}} = \{(q_i, r_i, a_i, G_i)\}$.

**Prompt for EG Extraction**

You are an expert medical knowledge engineer. Your task is to analyze a medical reasoning process and convert it into a highly structured and refined knowledge graph. You must extract, standardize, and logically connect all key medical concepts to produce a final, clean JSON output.

### Processing Rules:
#### 1. **Entity Construction: Standardization Purity (CRITICAL)**
1.1. **Extract Standardize**: Identify all medical entities (diseases, symptoms, tests, drugs, biomarkers). Immediately unify all synonymous, near-synonymous, or abbreviated entities (e.g., "the tumor", "the patient's mass") into the **single most medically precise and complete standard name** found in the source text (e.g., "invasive ductal carcinoma").
1.2. **Enforce Purity**: Entities MUST be core nouns or noun phrases only. All modifiers (adjectives, states, locations) must be handled in one of two ways:
1.2.1. **Splitting (Preferred)**: Decompose a complex entity into multiple, atomic triplets.
- **INCORRECT**: '["patient", "has symptom", "hard mass in upper outer quadrant of left breast"]'
- **CORRECT**: '[["patient", "has symptom", "breast mass"], ["breast mass", "has texture", "hard"], ["breast mass", "has location", "upper outer quadrant of left breast"]]'
1.2.2. **Modifier Integration**: Move the modifier into the relationship phrase.
- **INCORRECT**: '["HER2/neu positivity", "associated with", "poor prognosis"]'
- **CORRECT**: '["HER2/neu receptor", "positivity is associated with", "poor prognosis"]'

#### 2. **Relationship Definition**
2.1 Use clear, directional verb phrases (e.g., "causes", "is treated with", "is diagnosed by", "is positive for").
2.2 Explicitly mark negative relationships (e.g., "rules out", "is not associated with").
2.3 Retain hypothetical relationships from the reasoning process.
2.4 Convert time information into medical temporal expressions (e.g., "acute", "chronic").
2.5 Quantify probabilistic statements with risk levels (e.g., "high risk of").

#### 3. **Bridging Inference for Logical Gaps**
3.1. Review the reasoning flow to ensure logical completeness.
3.2. Add missing, but clinically essential, procedural steps that connect key events. A common failure is jumping from a symptom to a diagnosis without stating the intervening test.
- **Example**: If the text implies a diagnosis was made from a mass, you must add bridging triplets like '["patient", "undergoes", "biopsy"]' and '["biopsy", "was performed on", "mass"]'.

#### 4. **Output Format**
Your output must be a single JSON object with key: "triplets"

### Correct Output Example
{{ "triplets": [ ["patient", "has age", "65 years"], ["patient", "has gender", "female"], ["patient", "has symptom", "mass"], ["mass", "has texture", "hard"], ["mass", "is", "palpable"], ["mass", "located in", "left breast"], ["patient", "undergoes", "biopsy"], ["biopsy", "was performed on", "mass"], ["biopsy", "confirms diagnosis of", "invasive ductal carcinoma"], ["invasive ductal carcinoma", "is positive for", "HER2/neu receptor"], ["HER2/neu receptor", "positivity predicts", "a poor prognosis"], ["invasive ductal carcinoma", "is treated with", "Trastuzumab"] ] }}

### Reasoning process
{{rationale}}

## C.3 Details of Critical Evidence Graph Extraction

The process aims to refine a broad Evidence Graph, denoted as $\mathcal{G}$, into a concise **Critical Evidence Graph** ($G^*$). This $G^*$ is designed to encapsulate the most essential, step-by-step logical pathway required to reach a conclusion. The extraction pipeline involves three main stages: identifying a semantically relevant conclusion, extracting a causally connected subgraph, and refining this subgraph via transitive reduction.

The process begins not with graph topology, but with semantics. The **conclusion node** is identified from the set of all vertices $\mathcal{V}$ in the graph $\mathcal{G}$. This is achieved by employing a large language model, specifically *GPT-OSS-120B*, to find the single vertex that exhibits the maximal semantic similarity to the ground-truth answer $a$. This step anchors the entire subsequent extraction process to the correct final output.

Once the conclusion node is established, a **backward traversal** is performed starting from this node. This traversal explores the graph in reverse, collecting all parent nodes and their corresponding edges that form a path leading to the conclusion. The result is a complete, causally connected subgraph that contains all potential reasoning lines—direct and indirect—that culminate in the identified conclusion.

The final and most critical step is the refinement of this subgraph through **transitive reduction**. This procedure systematically prunes inferential shortcuts to enforce logical granularity. For any three nodes $A, B, C$ where a multi-hop path exists (e.g., $A \rightarrow B \rightarrow C$), any corresponding direct edge that "shortcuts" this path (i.e., $A \rightarrow C$) is removed. This ensures that every inferential step is made explicit, yielding a graph that represents the most parsimonious yet logically complete reasoning chain. The resulting pruned graph is the Critical Evidence Graph, $G^*$.

Finally, these generated graphs are compiled into the dataset. For each instance $i$, the original question $q_i$, reasoning $r_i$, and answer $a_i$ are aggregated with both the initial, broad Evidence Graph $G_i$ and the refined Critical Evidence Graph $G_i^*$. This forms the final data tuple $(q_i, r_i, a_i, G_i, G_i^*)$, and the collection of all such tuples comprises the complete dataset $\mathcal{D}_{\text{CEG}}$.

---

**Algorithm 1** Identify Conclusion and Extract Causal Subgraph

---

**Require:** Evidence Graph $\mathcal{G} = (\mathcal{V}, \mathcal{E})$; Ground-truth answer $a$; Semantic similarity model $\mathcal{M}$ (*GPT-OSS-120B*)
**Ensure:** Initial causal subgraph $G_{\text{sub}} = (\mathcal{V}_{\text{sub}}, \mathcal{E}_{\text{sub}})$
1: $v_c \leftarrow \arg\max_{v \in \mathcal{V}} \mathcal{M}_{\text{sim}}(v, a)$            ▷ Identify conclusion node
2: $\mathcal{V}_{\text{sub}} \leftarrow \{v_c\}$
3: $\mathcal{E}_{\text{sub}} \leftarrow \emptyset$
4: $Q \leftarrow [v_c]$            ▷ Initialize queue for backward traversal
5: visited $\leftarrow \{v_c\}$
6: **while** $Q$ is not empty **do**
7:      $u \leftarrow Q.\text{dequeue}()$
8:      **for** each vertex $p$ such that $(p, u) \in \mathcal{E}$ **do**            ▷ Find all predecessors of $u$
9:          **if** $p \notin$ visited **then**
10:             visited.add($p$)
11:             $Q.\text{enqueue}(p)$
12:             $\mathcal{V}_{\text{sub}}.\text{add}(p)$
13:          **end if**
14:          $\mathcal{E}_{\text{sub}}.\text{add}((p, u))$            ▷ Add edge to the subgraph
15:      **end for**
16: **end while**
17: $G_{\text{sub}} \leftarrow (\mathcal{V}_{\text{sub}}, \mathcal{E}_{\text{sub}})$
18: **return** $G_{\text{sub}}$

---

---

**Algorithm 2** Refine Subgraph via Transitive Reduction

---

**Require:** Causal subgraph $G_{\text{sub}} = (\mathcal{V}_{\text{sub}}, \mathcal{E}_{\text{sub}})$
**Ensure:** Critical Evidence Graph $G^* = (\mathcal{V}_{\text{sub}}, \mathcal{E}^*)$
1: $\mathcal{E}^* \leftarrow \mathcal{E}_{\text{sub}}$ ▷ Initialize edge set for the final graph
2: **for** each edge $(u, w) \in \mathcal{E}_{\text{sub}}$ **do**
3:     $G'_{\text{temp}} \leftarrow (\mathcal{V}_{\text{sub}}, \mathcal{E}_{\text{sub}} \setminus \{(u, w)\})$ ▷ Temporarily remove edge
4:     **if** PathExists$(u, w, G'_{\text{temp}})$ **then** ▷ Check for an alternative path from $u$ to $w$
5:         $\mathcal{E}^* \leftarrow \mathcal{E}^* \setminus \{(u, w)\}$ ▷ Prune shortcut edge if path exists
6:     **end if**
7: **end for**
8: $G^* \leftarrow (\mathcal{V}_{\text{sub}}, \mathcal{E}^*)$
9: **return** $G^*$

---

# D DETAILS OF PROGRESSIVE LEARNING PARADIGM

---

**Algorithm 3** Progressive Learning for Clinical Reasoning

---

**Require:** Base Model $M_{\text{base}}$, $\mathcal{D}_{\text{cold}} = \{(q, a, G)\}$, $\mathcal{D}_{\text{rl}} = \{(q, a, G^*)\}$, and $R_{\text{CRP}}$
**Ensure:** Final Clinically Reasoning Model $M_{\text{reason}}$
1: **Stage 1: Cold Start**
2: Initialize $M_{\text{cold}} \leftarrow M_{\text{base}}$
3: **for** minibatch $\{(q_b, a_b, G_b)\}_{b=1}^B \sim \mathcal{D}_{\text{cold}}$ **do**
4:     $y_b \leftarrow \text{Concat}(\textit{GPT-OSS-120B}(G_b), a_b)$
5:     Calculate $\mathcal{L}_{\text{SFT}}$ using Eq.(6) with $q_b$ and $y_b$
6:     UpdateParameters$(M_{\text{cold}}, \nabla \mathcal{L}_{\text{SFT}})$
7: **end for**
8: **Stage 2: Reinforcement Learning**
9: **for** batch $\{(q_n, a_n, G_n^*)_{n=1}^N\}_{b=1}^B \sim \mathcal{D}_{\text{rl}}$ **do**
10:     Initialize $\pi_\theta \leftarrow M_{\text{cold}}$, $\pi_{\text{ref}} \leftarrow M_{\text{cold}}$
11:     $\pi_{\theta_{\text{old}}} \leftarrow \pi_\theta$
12:     Initialize experience buffer $\mathcal{B}$
13:     **for** $(q_n, a_n, G_n^*) \in \text{batch}_b$ **do**
14:         generated_reasoning$_n \sim \pi_{\theta_{\text{old}}}(\cdot \mid q_n)$
15:         Calculate $R_{\text{CRP}_n}$ using CRP Reward with $q_n$, $G_n^*$, and $a_n$
16:         $\mathcal{B}.\text{add}(q_n, \text{generated\_reasoning}_n, R_{\text{CRP}_n})$
17:     **end for**
18:     **for** $k = 1, \ldots, K_{\text{opt}}$ **do**
19:         Calculate $\hat{A}_t$ using $R_{\text{CRP}}$ for all trajectories in $\mathcal{B}$
20:         $\mathcal{J}_{\text{GRPO}} \leftarrow \mathbb{E}[\text{clipped\_surrogate\_objective}(\hat{A}_t) - \beta \cdot \mathbb{D}_{KL}(\pi_\theta || \pi_{\text{ref}})]$ in Eq.(5)
21:         UpdateParameters$(\pi_\theta, \nabla \mathcal{J}_{\text{GRPO}})$
22:     **end for**
23: **end for**
24: $M_{\text{final}} \leftarrow \pi_\theta$
25: **return** $M_{\text{final}}$

---

## D.1 COLD START

The first stage of our training paradigm focuses on equipping the model with foundational knowledge of clinical reasoning pathways through supervised fine-tuning (SFT). To bridge the gap between structured reasoning graphs and sequential text generation, we employ *GPT-OSS-120B* to systematically transform each evidence graph $G$ into a coherent natural language reasoning sequence $y$ that preserves logical dependencies and clinical precision. This process enables the model to learn from structured medical reasoning procedure while maintaining its natural language generation capabilities. The training objective minimizes the standard cross-entropy loss over the transformed sequences, compelling the model to learn the mapping from a clinical querie $x$ to detailed reasoning

processes $y$:

$$\mathcal{L}_{\text{SFT}} = - \sum_{(x_i, y_i) \in \mathcal{D}_{\text{sft}}} \sum_{t=1}^{|y_i|} \log P_\theta(y_{i,t} | x_i, y_{i,<t}) \tag{6}$$

This SFT phase ensures the model masters the vocabulary, relational patterns, and overall structure of valid clinical arguments, forming a robust foundation for the subsequent reinforcement learning stage.

## D.2   DETAILS OF THINKING PROCESS REWRITING

To ensure the generated rationales maintain natural language flow while preserving the integrity of the underlying inferential logic, we employ *GPT-OSS-120B* to transform each reasoning graph $G$ into a logically coherent reasoning sequences $s_{\text{r}}$ in natural language through prompt below.

---

**Prompt for Thinking Process Rewriting**

Based on the original question and the reasoning graph, please rewrite the rationale to generate a standardized thought process.

Original Question: {{Question}}
Rationale: {{Rationale}}
Reasoning Graph: {{Reasoning Graph}}
Thinking Process Example: {{Thinking Process Example}}

Please provide the rewritten thinking process directly.

---

**Experimental Validation of Reasoning Graph-Guided Rewriting**   We conducted a validation experiment to assess the efficacy of using Reasoning Graphs for rewriting rationales. Our findings reveal that directly rewriting from a rationale leads to a loss of inferential information, specifically the omission of key reasoning nodes. We demonstrate that this issue can be effectively mitigated by including the reasoning graph in the prompt, which maximizes the preservation of the underlying logical structure. Specifically, using a random sample of 1,000 cases, we leveraged *GPT-OSS-120B* to rewrite rationales with and without the corresponding graphs. By extracting new reasoning graphs from the rewritten outputs and measuring their Jaccard Similarity to the originals, we found a significant improvement: the average similarity rose from 0.71 to 0.92 when the graph was provided.

## D.3   REWARD MODEL IMPLEMENTATION DETAILS

This section elaborates on the implementation details of the three core reward used in our reasoning process reward model ($R_{\text{reason}}$): Node Coverage ($R_{\text{node}}$), Structural Correctness ($R_{\text{struct}}$), and Reasoning Chain Completeness ($R_{\text{chain}}$). For each reward, we provide a textual description and pseudocode to bridge the gap between the theoretical formulas and the practical computation.

**Preprocessing: Semantic Element Mapping**   Before comparing the generated graph $\tilde{G}$ with the ground-truth graph $G^*$, a crucial preprocessing step is to establish a semantic mapping between the elements (nodes and relations) of the two graphs. Since the phrasing of entities or relations generated by the model may differ from the ground truth, we employ a "soft" matching based on semantic similarity rather than strict string matching.

The process is as follows:

1. **Element Collection and Embedding**: All unique elements from both $\tilde{G}$ and $G^*$ are extracted. An embedding model (e.g., *bge-large-en-v1.5*) is then called to generate high-dimensional vector representations for each element.

2. **Similarity Calculation**: A cosine similarity matrix $\mathbf{S}$ is computed between the elements of the generated graph and the ground-truth graph, where $\mathbf{S}_{ij} = \text{sim}(\tilde{e}_i, e_j^*)$.

3. **Map Establishment**: Based on preset thresholds for entities ($\theta_{\text{entity}}$) and relations ($\theta_{\text{relation}}$), we construct a mapping $\mathcal{M}$. for each element $e^*$ in $G^*$, this map contains all elements $\tilde{e}$ from $\tilde{G}$ whose similarity score with $e^*$ exceeds the corresponding threshold.

**Node Coverage** ($R_{\textbf{node}}$)   This reward evaluates whether the generated reasoning process incorporates all essential concepts. It is calculated by taking the average of the maximum similarity scores between each node in the ground-truth graph and all nodes in the generated graph.

$$R_{\text{node}} = \frac{1}{|\mathcal{V}^*|} \sum_{u \in \mathcal{V}^*} \max_{v \in \tilde{\mathcal{V}}} \{\text{sim}(u, v)\}$$

**Structural Correctness** ($R_{\textbf{struct}}$)   This metric assesses the factual accuracy of the reasoning relationships by calculating the proportion of correctly "recalled" ground-truth triplets. In our implementation, a ground-truth triplet $t^* = (s^*, p^*, o^*) \in G_{\text{c}}{}^*$ is considered "recalled" if there exists at least one triplet $(\tilde{s}, \tilde{p}, \tilde{o})$ in the generated graph $\tilde{G}_{\text{r}}$ where $\tilde{s}, \tilde{p}, \tilde{o}$ are valid semantic mappings of $s^*, p^*, o^*$, respectively.

$$R_{\text{struct}} = \frac{1}{|G_{\text{c}}{}^*|} \sum_{t^* \in G_{\text{c}}{}^*} \mathbb{I}(t^* \in \tilde{G}_{\text{r}}).$$

**Reasoning Chain Completeness** ($R_{\textbf{chain}}$)   This metric assesses logical coherence by measuring the proportion of ground-truth triplets that form a single, unbroken line of reasoning. The calculation first identifies all successfully recalled ground-truth triplets. An undirected graph is then constructed from these triplets, and its largest connected component is found. The final score is the ratio of the number of triplets within this largest component to the total number of ground-truth triplets.

$$R_{\text{chain}} = \frac{|C_{\max} \cap G_{\text{c}}{}^*|}{|G_{\text{c}}{}^*|}$$

# E   DETAILS OF EVALUATION

## E.1   BENCHMARK INTRODUCTION

This section provides a detailed description of the six benchmarks used for evaluating our model, categorized as in-domain and out-of-domain.

**In-Domain Benchmarks:**

- **MedQA** (Jin et al., 2021) is a large-scale, multiple-choice question answering dataset based on the United States Medical Licensing Examination (USMLE). It is designed to evaluate clinical knowledge and reasoning on a wide range of medical topics. The dataset is available in several languages and formats, testing a model's ability to apply extensive medical knowledge to solve challenging problems.

- **MedBullets-5op** (Chen et al., 2025) is a benchmark derived from the MedBullets medical education platform, which provides multiple-choice questions aligned with the USMLE Step 1, 2, and 3 exams. It covers a comprehensive curriculum spanning basic science, clinical knowledge, and patient management. Our evaluation utilizes 5-option (MedBullets-5op) versions to assess performance on varying levels of question difficulty.

- **MedCase** (Wu et al., 2025b) is a dataset specifically designed to test multi-step clinical diagnostic reasoning. It consists of open-ended questions formulated from complex, real-world clinical case reports. This benchmark challenges a model's ability to synthesize patient information, follow a logical diagnostic workflow, and understand nuanced clinical narratives.

**Out-of-Domain Benchmarks:**

- **MMLU-H (Massive Multitask Language Understanding)** (Hendrycks et al., 2020) is a subset of is MMLU, a broad benchmark designed to measure knowledge across 57 diverse subjects. For our evaluation, we utilize the health and medical-focused subsets, such

as "Clinical Knowledge", "College Medicine", "Professional Medicine", "Medical Genetics", "College Biology", "Anatomy" . These subsets test for expert-level knowledge in specialized medical or biology domains.

- **MMLU-Pro-H** (Wang et al., 2024) is a subset of MMLU-Pro, an advanced and more robust version of the original MMLU. It was developed to address limitations such as question ambiguity by having domain experts rewrite and validate the questions. MMLU-Pro provides a more reliable and difficult assessment of a model's expert-level reasoning capabilities. We use its corresponding health-focused subsets for evaluation.

- **DiagArena** (Zhu et al., 2025) is a challenging benchmark designed to rigorously evaluate the diagnostic reasoning capabilities of large language models in scenarios that mirror the complexity of real-world clinical practice. Developed to overcome the limitations of existing medical benchmarks, which often rely on simplified, multiple-choice questions, DiagnosisArena provides a more authentic assessment of a model's ability to perform professional-level diagnostic reasoning. The benchmark is composed of 1,113 challenging patient cases sourced from clinical reports in 10 top-tier medical journals, including The Lancet and NEJM. This ensures the cases are authentic, complex, and diverse, spanning 28 different medical specialties.

### E.2 OPEN-ENDED QUESTION JUDGEMENT

We consistently evaluate the open-ended questions through *GPT-OSS-120B*, using the prompt below.

---

**Prompt for Open-ended Question Judgement**

Is our predicted answer correct? [yes/no]
Predicted answer: {{response_answer}}
Actual answer: {{golden_answer}}

---

### E.3 CLINICAL REASONING QUALITY ASSESSMENT

To establish a standardized and quantifiable methodology for the evaluation of clinical reasoning, we developed a comprehensive assessment framework. This framework is engineered to promote and rigorously appraise structured, transparent, and evidence-based clinical thinking. Its intended applications are manifold, encompassing the formative and summative assessment of medical students and resident physicians, performance validation of diagnostic models, quality assurance for Case-Based Discussions, and the retrospective analysis for continuous improvement of clinical decision-making.

We implemented this framework by employing a committee of three distinct large language models—namely, *DeepSeek-R1*, *Qwen3-235B*, and *Gemini-2.5-Pro*—to ensure a robust and multifaceted assessment. This committee evaluates the quality of the reasoning process against five core criteria. This multi-dimensional approach aligns with best practices, such as scoring each dimension independently to mitigate the "halo effect." The five criteria are as follows:

- **Logical Coherence:** This criterion evaluates the internal consistency and deductive validity of the clinical reasoning process. The assessment focuses on whether the thought process unfolds in a structured and rational manner, moving from evidence to conclusion without logical fallacies or contradictions. A coherent argument should demonstrate a clear chain of reasoning where each step justifiably follows from the previous one.

- **Factual Accuracy:** This dimension verifies the correctness of the medical knowledge that underpins the reasoning. It ensures that the argument is built upon a foundation of established clinical facts, principles, and data. This includes the accurate application of pathophysiology, epidemiology, diagnostic criteria, and understanding of clinical signs and symptoms.

- **Evidence Faithfulness:** This criterion assesses whether the reasoning is strictly grounded in the information provided in the case. The conclusions and intermediate hypotheses must be directly supported by the available evidence (e.g., patient history, physical examination

findings, and diagnostic results) without fabricating or "hallucinating" information. This dimension is crucial for ensuring that the reasoning process does not stray from the specific context of the case by making assumptions or introducing external data that was not provided.

- **Interpretability & Clarity:** This evaluates the transparency and comprehensibility of the articulated reasoning. The goal is to determine if a human clinical expert can easily follow the thought process from beginning to end. High-quality reasoning should be presented in a clear, organized, and unambiguous manner, using precise medical terminology correctly. It should avoid convoluted language or poorly structured arguments that obscure the underlying logic. This criterion measures not just the quality of the thinking itself, but also the quality of its communication, ensuring the rationale is transparent and open to scrutiny.

- **Information Utilization:** This dimension measures the thoroughness with which all relevant information from the case is considered and integrated into the final analysis. It assesses whether the reasoning effectively incorporates both key positive findings that support a diagnosis and pertinent negative findings that help rule out others. A high-scoring evaluation would demonstrate that no significant piece of data has been overlooked.

**Standardized Reference Case** To make the assessment criteria concrete, this framework uses the following standardized case as the basis for all dimensional examples. Accordingly, we designed a standardized case and built the following evaluation examples around it.

---

**Case Information**

**Case Summary:** A 55-year-old male with a long history of smoking and hypertension presents to the emergency department with a two-hour history of "sudden-onset, retrosternal crushing pain". The pain radiates to his left arm and is accompanied by diaphoresis. Physical examination is unremarkable. An electrocardiogram (ECG) shows ST-segment elevation in leads V2-V4. The patient **denies** that the pain is related to breathing (non-pleuritic) and has **no** fever or unilateral leg swelling.
**Most Likely Diagnosis:** Acute Anterior Myocardial Infarction (AMI).
**Key Differential Diagnoses:** Pulmonary Embolism (PE), Aortic Dissection.

---

**Five Assessment Dimensions**    We design the prompts to operationalize each of the five assessment dimensions, providing the evaluating models with a core definition and a structured task.

---

### 1. Logical Coherence

**Core Definition:** Assesses whether the chain of reasoning from evidence (case information) to conclusion (diagnosis) is complete, sound, and free of contradictions, and whether the final conclusion follows naturally and necessarily from the reasoning process.

**Score 2 (Excellent):** The reasoning chain is logically seamless, and the conclusion is a direct, necessary result of the evidence. The entire argument is solid and convincing from premise to conclusion, with no logical leaps or internal contradictions.

*Example:* "The patient presents with typical ischemic chest pain (crushing, radiating to the left arm, with diaphoresis). The key evidence is the ST-segment elevation in leads V2-V4, which directly localizes and confirms an acute anterior myocardial infarction. Therefore, the final diagnosis is AMI."

**Score 1 (Adequate):** The reasoning process generally supports the conclusion, but contains minor logical flaws. This may include insufficient justification for secondary points, minor inferential gaps, or a conclusion that is too broad/narrow to be precisely supported by the evidence.

*Example:* "The patient has chest pain and ECG abnormalities, indicating a cardiac issue. The ECG changes are consistent with cardiac ischemia. Therefore, the conclusion is Acute Coronary Syndrome (ACS)."

**Score 0 (Inadequate):** The reasoning process severely contradicts the conclusion, or the reasoning chain contains fundamental logical fallacies. The final answer appears random or incorrect, being completely disconnected from or contradictory to the analysis.

*Example:* "The ST-segment elevation on the ECG strongly suggests AMI. The absence of pleuritic pain also lowers the likelihood of a pulmonary embolism. Therefore, the final diagnosis is pulmonary embolism."

---

## 2. Factual Accuracy

**Core Definition:** Assesses the accuracy of all medical knowledge cited in the reasoning process, ensuring it aligns with current, evidence-based clinical guidelines, textbooks, and consensus.

**Score 2 (Excellent):** All cited medical facts are accurate and current. This includes disease definitions, pathophysiology, diagnostic criteria, interpretation of findings, and treatment principles, all consistent with authoritative sources.

*Example:* "The ECG shows ST-segment elevation in leads V2-V4, a hallmark feature of an acute anterior wall myocardial infarction, which is typically associated with the occlusion of the Left Anterior Descending (LAD) coronary artery"

**Score 1 (Adequate):** Contains non-critical factual errors that do not alter the main diagnostic pathway. For instance, citing a slightly inaccurate statistic or a minor error in a non-essential value range.

*Example:* "The ECG shows ST-segment elevation in leads V2-V4, which is indicative of an acute inferior wall myocardial infarction."

**Score 0 (Inadequate):** Contains one or more critical factual errors that fundamentally mislead the reasoning process or could lead to patient harm.

*Example:* "ST-segment elevation is a benign early repolarization pattern, common in healthy individuals, and therefore has no clinical significance in this case."

## 3. Evidence Faithfulness

**Core Definition:** Assesses whether the reasoning is strictly and exclusively based on the information provided in the case, avoiding any fabrication of data (i.e., "hallucination").

**Score 2 (Excellent):** Every step of the reasoning is explicitly traceable to specific information within the input case. All arguments are directly cited from or based on the source text, with no extrapolation beyond the given evidence.

*Example:* "Based on the case description of 'retrosternal crushing pain','radiating to the left arm', and 'accompanied by diaphoresis', an acute cardiac event is highly suspected."

**Score 1 (Adequate):** The reasoning is primarily based on case information but includes minor, reasonable clinical assumptions or slight misinterpretations of the evidence. It introduces small, clinically plausible details not explicitly stated in the text.

*Example:* "The patient's chest pain, likely accompanied by shortness of breath, points towards an acute cardiac event."

**Score 0 (Inadequate):** The reasoning contains clear "hallucinations", fabricating key information not present in the case and using it as a central pillar for the argument.

*Example:* "Laboratory results show the patient's troponin level is elevated at 15 ng/mL, confirming myocardial necrosis."

### 4. Interpretability & Clarity

**Core Definition:** Assesses whether the reasoning is presented in a structured, professional, and concise manner that is easily understood by a clinical peer.

**Score 2 (Excellent):** The presentation is clear, well-structured, and uses professional, precise language. It follows a standard clinical logic flow (e.g., presentation → differentials → analysis → conclusion), allowing a peer to effortlessly follow the complete thought process.

*Example:* "1. Clinical Presentation: The patient's symptoms (crushing chest pain, radiation, diaphoresis) are highly suggestive of cardiac-origin pain. 2. Key Investigations: The ECG finding of ST-segment elevation is definitive evidence for AMI. 3. Conclusion: Integrating the clinical picture and ECG, the diagnosis is clearly AMI."

**Score 1 (Adequate):** The core idea is understandable, but the presentation is flawed by redundancy, repetition, disorganized structure, or ambiguous language. It requires extra effort from the reader to parse the logic.

*Example:* "The patient has chest pain, very painful, and the EKG is also not good, it has changes. So we think it's a heart problem, because the pain and the EKG both point to the heart. So it should be a heart attack."

**Score 0 (Inadequate):** The presentation is convoluted, lacks a logical structure, and is filled with meaningless jargon or inappropriate terminology, making the core reasoning difficult or impossible to understand.

*Example:* "Vectorial changes of myocardial repolarization confirm the electrophysiological basis of transmural ischemia. Therefore, despite the negative evidence of pleuritic pain, the differential of dissection persists. The etiology is thus attributed to a cardiac source, an MI is considered."

### 5. Information Utilization

**Core Definition:** Assesses how thoroughly all key clinical information was utilized, especially how both positive findings and important **pertinent negatives** were integrated to form the final judgment.

**Score 2 (Excellent):** Comprehensively considers all diagnostically significant positive and negative findings. It not only identifies evidence supporting the conclusion but also explicitly explains how key negative findings help to rule out other relevant diagnoses.

*Example:* "The diagnosis of AMI is based not only on positive findings like crushing chest pain and ST elevation, but is also supported by pertinent negatives: the patient's denial of pleuritic pain and absence of leg swelling significantly lower the probability of other fatal causes like pulmonary embolism."

**Score 1 (Adequate):** Focuses on the most critical clinical information to support the conclusion but overlooks some secondary or diagnostically valuable clues (positive or negative). The analysis is not fully comprehensive.

*Example:* "The patient's crushing chest pain and ST-segment elevation on the ECG are classic signs of an acute myocardial infarction."

**Score 0 (Inadequate):** Exhibits clear "cherry-picking" behavior. It selectively focuses on evidence that supports a preconceived conclusion while systematically ignoring critical information or pertinent negatives that contradict it.

*Example:* (*Assuming the ECG in this case was normal*) "The patient's chest pain is classic crushing, radiating pain, therefore the diagnosis is acute myocardial infarction."

# F  IMPLEMENTATION DETAILS

**Cold Start**

The supervised fine-tuning (SFT) phase was implemented utilizing the *LLaMA-Factory* (Zheng et al., 2024) framework. We conducted full-parameter fine-tuning on the *Llama-3.1-8B-Instruct*. The optimization was performed for 8 epochs, employing a cosine learning rate decay schedule initialized from a peak learning rate of $1E-6$ with a warmup proportion of $0.1$. An effective batch size of 2 was maintained by configuring a per-device batch size of 1 and accumulating gradients over 2 steps. To enhance computational efficiency, we leveraged bfloat16 mixed-precision training and adopted the DeepSpeed ZeRO-2 strategy for distributed execution. This stage was conducted on 8xH100 GPUs and took 1 hour. A comprehensive summary of the SFT hyperparameters is provided in Table 3.

Table 3: Hyperparameters for the Supervised Fine-Tuning stage.

| Parameter | Value |
|---|---|
| *Model Configuration* | |
| Base Model | `Llama-3.1-8B-Instruct` |
| Finetuning Strategy | Full-parameter |
| *Optimization Parameters* | |
| Learning Rate | 1E-6 |
| LR Scheduler | Cosine Annealing |
| Warmup Proportion | 0.1 |
| Training Epochs | 8.0 |
| Per-Device Batch Size | 1 |
| Gradient Accumulation Steps | 2 |
| Optimizer Precision | BF16 |
| Distributed Framework | DeepSpeed (ZeRO Stage 2) |
| *Data Handling* | |
| Max Sequence Length | 10,240 |

**Reinforcement Learning**

The subsequent reinforcement learning (RL) stage was executed using the *VERL* (Sheng et al., 2024) framework, which facilitates our Graph-based Reward Policy Optimization (GRPO) algorithm. Both the actor and the reference policies were initialized from the converged parameters of the model obtained in the Cold Start stage.

The composite Clinical Reasoning Procedure (CRP) reward signal was constructed with the following weights: $w_{\text{reason}} = 0.3$, $w_{\text{answer}} = 0.6$, $w_{\text{format}} = 0.1$. The internal components of the reasoning reward, $R_{\text{reason}}$, were weighted as $\lambda_{\text{node}} = 0.5$, $\lambda_{\text{struct}} = 0.3$, and $\lambda_{\text{chain}} = 0.2$. These hyper-parameters were determined empirically through preliminary validation experiments. Vector representations of clinical concepts, necessary for the $R_{\text{node}}$ calculation, were generated using the *bge-large-en-v1.5* embedding model.

The actor policy was optimized over 10 epochs. The learning rate was set to $5E-7$ with a warmup proportion of $0.185$. To maintain training stability and prevent significant policy deviation from the SFT initialization, we incorporated a KL-divergence penalty with a coefficient $\beta = 0.001$ against the reference policy. The GRPO update step was configured with a global batch size of 32, a mini-batch size of 16, and a per-GPU micro-batch size of 4. During the experience generation (rollout) phase, 8 responses were sampled for each input prompt at a temperature of $1.0$. For evaluation, responses were generated deterministically (temperature=0.0). The distributed training process was conducted on a single compute node provisioned with 8xH100 GPUs for 3 days. Detailed hyperparameters for the RL stage are enumerated in Table 4.

Table 4: Hyperparameters for the Reinforcement Learning stage.

| Parameter | Value |
|---|---|
| *Model & Algorithm* | |
| Policy Initialization | Cold Start Model |
| RL Algorithm | GRPO (via `VERL`) |
| KL Penalty Coefficient ($\beta$) | 0.001 |
| *Optimization Parameters* | |
| Actor Learning Rate | $5.0 \times 10^{-7}$ |
| LR Warmup Proportion | 0.185 |
| Total Epochs | 10 |
| Global Batch Size | 32 |
| PPO Mini-batch Size | 16 |
| Per-GPU Micro-batch Size | 4 |
| *Rollout Configuration* | |
| Generations per Prompt ($n$) | 8 |
| Rollout Temperature | 1.0 |
| Evaluation Temperature | 0.0 |
| Max Prompt Length | 4096 |
| Max Response Length | 10240 |
| *Reward Function Weights* | |
| $w_{\text{reason}}, w_{\text{answer}}, w_{\text{format}}$ | 0.3, 0.6, 0.1 |
| $\lambda_{\text{node}}, \lambda_{\text{struct}}, \lambda_{\text{chain}}$ | 0.5, 0.3, 0.2 |

# G  CASE STUDY

In this section, we present a comparative analysis of the reasoning processes for 3 models: *MedS³-8B*, *MedReason-8B*, and our own model, MedCEG. To ensure a fair comparison, we selected a case study where all models arrived at the correct conclusion. For each model, we will display its step-by-step reasoning, followed by a human expert's evaluation, score, and detailed justification for that score.

**Question**

**Case Information:**
A man in his 20s presented with thickening and swelling of hands after immersion in water, accompanied by burning pruritus for 3 years. The lesions occurred as symmetrical white and transparent papules, white keratosis, excessive wrinkling on the dorsum of hands after water immersion and worsened in summer, and remitted in winter. He was treated with topical tretinoin ointment intermittently at a local clinic with little improvement. The lesions gradually aggravated, and the affected area dispersed to wrists and elbows during past 1.5 years. He attributed these changes to frequent hand washing during the COVID-19 epidemic. No family history of similar skin problems and no history of hyperhidrosis, cystic fibrosis, atopy, or trauma to the hands.

**Physical Examination:**
After immersing both hands in water for 10 mins, there were symmetrical white papules, white hyperkeratosis spots, and excessive wrinkling on the thenar eminence, wrist flexure of both hands, and dorsum of fingers and hands.

**Diagnostic Tests:**

- **Skin Biopsy:** Histopathologic examination revealed remarkable compact orthohyperkeratosis in the epidermis, dilatation of intraepidermal eccrine ducts, and hyperplasia of the eccrine sweat glands.

- **Images:**
  - *Image Title:* Clinical presentation after water immersion
    *Image Description:* A and B - Symmetrical white papules, white hyperkeratosis spots, and excessive wrinkling on hands after water immersion.
  - *Image Title:* Histopathologic features
    *Image Description:* C and D - Hyperkeratosis, dilated eccrine ducts in the epidermis and dermis, and hyperplasia of the eccrine sweat gland (hematoxylin-eosin staining).

**Here are the four options:**
A: Aquagenic syringeal acrokeratoderma (ASA)
B: Aquagenic keratoderma
C: Aquagenic palmoplantar keratoderma
D: Aquagenic wrinkling of the palms

> **_AlphaMed-8B_'s Original Response**
>
> Let's analyze the information provided step by step.
>
> 1. **Patient Presentation**: The patient is a young man who presents with thickening and swelling of his hands after immersion in water, accompanied by burning pruritus. He has symmetrical white and transparent papules, white keratosis, and excessive wrinkling on the dorsum of his hands.
>
> 2. **Symptoms**: The key symptoms include symmetrical white papules, hyperkeratosis, and excessive wrinkling of the hands after water immersion. He experiences burning pruritus, particularly in summer, which improves in winter.
>
> 3. **Diagnosis**: The patient's condition is known as aquagenic syringeal acrokeratoderma (ASA). This condition involves symmetrical white papules, hyperkeratosis, and wrinkling of the hands after exposure to water.
>
> 4. **Diagnostic Tests**: A skin biopsy revealed compact orthohyperkeratosis, dilatation of intraepidermal eccrine ducts, and hyperplasia of the eccrine sweat glands.
>
> 5. **Options Provided**:
>    A. Aquagenic syringeal acrokeratoderma (ASA)
>    B. Aquagenic keratoderma
>    C. Aquagenic palmoplantar keratoderma
>    D. Aquagenic wrinkling of the palms
>
> 6. **Conditions**:
>    - **Aquagenic Syringeal Acrokeratoderma (ASA)**: This condition involves keratoderma, wrinkling of the hands after water exposure, and eccrine gland abnormalities.
>    - **Aquagenic Keratoderma**: This is a condition characterized by keratoderma after water exposure, but it doesn't always involve eccrine gland abnormalities.
>    - **Aquagenic Palmoplantar Keratoderma**: This involves keratoderma on the palms and soles after water exposure.
>    - **Aquagenic Wrinking of the Palms**: This involves wrinkling of the palms after water exposure.
>
> 7. **Choice Analysis**:
>    - Aquagenic syringeal acrokeratoderma (ASA) involves keratoderma, wrinkling of the hands after water exposure, and eccrine gland abnormalities, which matches the patient's presentation.
>    - Aquagenic keratoderma refers to keratoderma after water exposure but doesn't specify eccrine gland abnormalities.
>    - Aquagenic palmoplantar keratoderma involves keratoderma on the palms and soles, which is not specified in this case.
>    - Aquagenic wrinkling of the palms involves wrinkling of the palms after water exposure but doesn't include keratoderma.
>
> 8. **Conclusion**: Based on the information provided, the patient's condition involves symmetrical white papules, hyperkeratosis, wrinkling of the hands after water immersion, and eccrine gland abnormalities. The most likely diagnosis is aquagenic syringeal acrokeratoderma (ASA).
>
> 9. **Final Answer**: The final answer is \boxed{A}.

# Scoring Rationale

**Evidence Faithfulness (Score: 2):** The model perfectly adheres to the provided case information. The description of the patient's presentation, symptoms, and diagnostic test results are all accurately extracted from the source text without any addition or fabrication of data.

**Logical Coherence (Score: 0):** The reasoning process is critically flawed due to circular logic. The model states the final diagnosis (ASA) in step 3, at the very beginning of its analysis, and then re-states it in the conclusion (step 8). The intermediate steps do not build a logical bridge from evidence to conclusion but instead assume the conclusion from the start.

**Factual Accuracy (Score: 0):** The definitions provided in step 5 are factually incorrect and misleading. Aquagenic Syringeal Acrokeratoderma (ASA) is a specific histopathological **variant** of Aquagenic Keratoderma, distinguished by the eccrine duct changes seen in this case. By presenting them as mutually exclusive conditions, the model creates a false dichotomy that undermines the entire differential diagnosis.

**Interpretability & Clarity (Score: 1):** While the core message is understandable, the structure is disorganized and highly redundant. Step 7 ("Choice Analysis") largely repeats the flawed definitions from step 6 without adding any new analytical insight, making the argument inefficient and difficult for a clinical peer to follow.

**Comprehensiveness of Information Utilization (Score: 1):** The model identifies the key positive findings but fails to use them effectively. It does not explicitly highlight that the histopathological evidence of **dilated eccrine ducts** is the single most important clue that elevates a general diagnosis of aquagenic keratoderma to the specific ASA variant, thus missing the core diagnostic step.

*Huatuo-o1-8B's* **Original Response**

THINKING

Alright, let's think about this. We've got a young man in his 20s who's been dealing with some pretty uncomfortable symptoms on his hands. He's got thickening and swelling, and it's been going on for three years. That's a long time to be dealing with this. The symptoms get worse when he's in water and improve when it's cold, which is interesting. That's a big clue.

Now, let's look at what's happening on his skin. He's got these symmetrical white papules and white patches, and his skin gets all wrinkly after he's been in water. That's not just any kind of skin issue. It's like his skin is reacting to the water in a specific way.

The biopsy results are really telling. There's compact hyperkeratosis in the epidermis, which means there's a thickening of the outer layer of the skin. And there's dilation of the eccrine ducts and hyperplasia of the eccrine sweat glands. That's a lot of changes happening in the skin's sweat glands. This is starting to sound like a specific condition.

Let's think about what this could be. Aquagenic syringeal acrokeratoderma, or ASA, is a condition that fits these symptoms. It's known for causing skin changes after water exposure, and it involves the eccrine glands. That's exactly what we're seeing here.

Now, let's compare this to the other options. Aquagenic keratoderma and aquagenic palmo-plantar keratoderma both involve skin thickening after water exposure, but they don't specifically mention the eccrine glands. And aquagenic wrinkling of the palms is more about the skin wrinkling without the other symptoms we're seeing.

So, putting it all together, the symptoms, the biopsy results, and the way the condition worsens with water exposure all point to aquagenic syringeal acrokeratoderma. It's the only one that matches everything we're seeing here. Yeah, that's definitely the one. The answer is A.

FINAL RESPONSE

The answer is \boxed{A}.

---

# Scoring Rationale

**Evidence Faithfulness (Score: 2):** The model demonstrates excellent faithfulness to the case details. Throughout its "Thinking" process, it accurately recalls the patient's history, clinical signs, and specific biopsy findings, using them as the sole basis for its reasoning without hallucinating any information.

**Logical Coherence (Score: 1):** The reasoning follows a generally logical path from evidence to conclusion. However, the argument contains a minor flaw in its differential analysis. While it correctly identifies that other options "don't specifically mention the eccrine glands," it doesn't fully articulate why the presence of dilated ducts is the definitive link to ASA, leaving a small logical gap in its final conclusion.

**Factual Accuracy (Score: 1):** The response avoids making a critical factual error but lacks precision. By stating that other aquagenic keratodermas "don't specifically mention the eccrine glands," it implies they are entirely separate conditions. The more accurate fact is that ASA is a specific clinicopathologic variant of Aquagenic Keratoderma, defined by those very gland changes. This nuance is missed, constituting a non-critical factual weakness.

**Interpretability & Clarity (Score: 1):** The entire "Thinking" section is written in a highly conversational and verbose style ("That's a long time to be dealing with this," "Yeah, that's definitely the one."). While the reasoning is understandable, this informal tone is inappropriate for professional clinical analysis and requires a peer to sift through colloquialisms to extract the core logic.

**Comprehensiveness of Information Utilization (Score: 1):** The model successfully uses most of the key information. However, it fails to make the most critical connection explicitly: it doesn't link the specific histopathologic finding of "dilatation of intraepidermal eccrine **ducts**" to the "**syringeal**" (relating to a tube or duct) component of the correct diagnosis. This omission prevents the analysis from being fully comprehensive.

---

**MedCEG's Original Response**

1. **Analyze the user's request:** The user has provided a clinical case study of a man in his 20s with specific skin symptoms related to water immersion. They've included details about the presentation, history, physical exam, and histopathology. The user wants me to identify the most likely diagnosis from a list of four options.

2. **Deconstruct the clinical presentation:**

   - **Patient:** Man in his 20s.
   - **Chief Complaint:** Thickening and swelling of hands after immersion in water, accompanied by burning pruritus.
   - **History:**
     - Duration: 3 years.
     - Triggers: Water immersion.
     - Symptoms: Symmetrical white and transparent papules, white keratosis, excessive wrinkling on the dorsum of hands.
     - Aggravating factor: Summer.
     - Relieving factor: Winter.
     - Previous treatment: Topical tretinoin with little improvement.
     - Progression: Gradually aggravated, spread to wrists and elbows over 1.5 years.
     - Associated factors: Attributed to frequent hand washing during the COVID-19 epidemic. No family history, no hyperhidrosis, cystic fibrosis, atopy, or trauma.
   - **Physical Examination:**
     - After 10 minutes of water immersion: Symmetrical white papules, white hyperkeratosis spots, excessive wrinkling on thenar eminence, wrist flexure, and dorsum of fingers and hands. This is the key clinical finding. The description "white papules" and "white hyperkeratosis" is a crucial clue.
   - **Diagnostic Tests:**
     - **Histopathology:** Remarkable compact orthohyperkeratosis in the epidermis, dilatation of intraepidermal eccrine ducts, and hyperplasia of the eccrine sweat glands. This is a very specific finding.

3. **Evaluate the options against the clinical presentation:**

   - **A. Aquagenic syringeal acrokeratoderma (ASA)**
     - **What is it?** A rare skin disorder characterized by acrokeratoderma (thickening of the outer layer of the skin) that appears after water immersion. It's also known as aquagenic wrinkling of the palms.
     - **Clinical features:** Symmetrical, white, translucent papules and plaques on the palms and sometimes the dorsum of the hands, appearing within minutes of water immersion. Excessive wrinkling is a hallmark. Pruritus can occur.
     - **Histopathology:** Hyperkeratosis, acanthosis, and dilatation of the eccrine ducts. This matches the provided histopathology.
     - **Fit with the case:**
       * Trigger: Water immersion - Yes.
       * Symptoms: White papules, hyperkeratosis, excessive wrinkling - Yes, this is a perfect match.
       * Location: Dorsum of hands, wrists - Yes.
       * Histopathology: Dilatation of eccrine ducts, hyperkeratosis - Yes, this is a perfect match.
       * The patient's attribution to frequent hand washing fits the context.
       * The progression to wrists and elbows is also possible.
       * The pruritus is also consistent.

**MedCEG's Original Response (Continuation)**

3. **Evaluate the options...** (Continued)

- **A. ... (ASA)** (Continued)
  - **Fit with the case:** (Continued)
    * This seems like a very strong candidate.
- **B. Aquagenic keratoderma**
  - **What is it?** A general term for skin thickening of the palms and soles that occurs after water immersion. It's a broader category.
  - **Fit with the case:** This is a possible, but less specific, diagnosis. The description "aquagenic syringeal acrokeratoderma" is more specific and fits the details better. The "syringeal" part points directly to the eccrine glands, which is confirmed by the histopathology. "Aquagenic keratoderma" is a good descriptor but not the most precise diagnosis given the specific features.
- **C. Aquagenic palmoplantar keratoderma**
  - **What is it?** A group of disorders characterized by thickening of the skin on the palms and soles after water immersion.
  - **Fit with the case:** The location (palms and soles) is a bit of a mismatch. The case describes involvement of the dorsum of hands, wrists, and elbows, not specifically the palms.
- **D. Aquagenic wrinkling of the palms**
  - **What is it?** This is another name for Aquagenic Syringeal Acrokeratoderma (ASA). It describes the primary feature of excessive wrinkling that occurs after water immersion on the palms.
  - **Fit with the case:** This is essentially the same entity as option A. The case describes more than just wrinkling; it describes white papules and hyperkeratosis. Therefore, A is a more complete description of the clinical picture.

4. **Synthesize and conclude:**

- The patient presents with a classic triad of symptoms: (1) water-induced skin changes on the hands, (2) white papules and hyperkeratosis, and (3) excessive wrinkling.
- The histopathology confirms the involvement of the eccrine sweat glands (dilatation of ducts, hyperplasia of glands), which is a key feature of this condition.
- The name "Aquagenic Syringeal Acrokeratoderma" perfectly encapsulates these features: "aqua-" (water), "-genic" (induced by), "-syringeal" (related to sweat glands), and "-acrokeratoderma" (skin thickening).
- The other options are either too general (B) or don't fit the specific clinical and histological details as well (C, D).

5. **Final check:**

- The case description is a textbook presentation of Aquagenic Syringeal Acrokeratoderma (ASA).
- The key features are all present: water-induced white papules, hyperkeratosis, and wrinkling, along with the characteristic histopathology of eccrine duct dilatation and hyperplasia.

FINAL RESPONSE

Based on the clinical presentation and histopathological findings, the final answer is \boxed{A}.

# Scoring Rationale

**Logical Coherence (Score: 2):** The model's reasoning is exceptionally clear and sound. It follows a perfect clinical reasoning pathway: deconstructing the case data, systematically evaluating each option against the evidence, and synthesizing the findings into a logical conclusion. There are no fallacies or gaps in the argument.

**Factual Accuracy (Score: 2):** All medical facts presented are accurate. The model correctly defines Aquagenic Syringeal Acrokeratoderma (ASA) and correctly identifies the relationship between the different diagnostic terms, noting that "Aquagenic keratoderma" is a broader category and "Aquagenic wrinkling of the palms" is a synonym for ASA. This demonstrates a precise understanding of the topic.

**Evidence Faithfulness (Score: 2):** The response is perfectly faithful to the source material. The "Deconstruct the clinical presentation" step is a masterclass in extracting and organizing information from the case without adding or fabricating any details, ensuring the entire analysis is grounded in the provided evidence.

**Interpretability & Clarity (Score: 2):** The response is exemplary in its clarity and structure. The use of a multi-step process ("Thinking" and "Final Response"), clear headings, and nested lists makes the entire thought process transparent and effortless for a clinical peer to follow. The final response is a concise and professional summary of the detailed analysis.

**Comprehensiveness of Information Utilization (Score: 2):** The model utilizes all key pieces of information to their full diagnostic potential. Crucially, it makes the explicit connection between the histopathology ("dilatation of... eccrine ducts") and the term "-syringeal" in the correct diagnosis, demonstrating a deep and comprehensive understanding of the pathophysiology and terminology.

