# OpenReview forum: "MedCEG: Reinforcing Verifiable Medical Reasoning with Critical Evidence Graph"
_ICLR.cc/2026/Conference — ICLR 2026 Conference Withdrawn Submission_

### Official Review · Reviewer_RYEM · 2025-11-01

**Soundness:** 2
**Presentation:** 3
**Contribution:** 2
**Rating:** 4
**Confidence:** 5

**Summary:**

This paper introduces MedCEG, a framework that enhances medical reasoning in large language models by supervising their reasoning process through a Critical Evidence Graph (CEG). The framework leverages multiple advanced LLMs to generate triplets and medical graphs, and subsequently improves smaller models’ performance on medical QA benchmarks such as MedQA.

**Strengths:**

1. The paper is well-structured and clearly written, making it easy to follow.

2. The focus on medical reasoning, aiming for both reliable reasoning processes and correct answers, is meaningful and relevant for advancing LLM performance on clinical tasks.

3. Combining reasoning chains and graph structures as a reward signal enhances both accuracy and reasoning quality, as evidenced by detailed results in several medical question datasets.

**Weaknesses:**

1. The entire pipeline heavily depends on several powerful LLMs for generating reasoning processes, constructing graphs, computing rewards, and even performing model evaluation (via LLM-as-a-judge). None of these steps involves natural linkages or expert verification, which raises concerns about cumulative noise and inherited bias.

2. The evaluation design is largely self-referential. GPT-OSS-120B is used to generate reasoning traces for SFT training in the cold-start stage, and the same family of models is later reused for process parsing, reward computation, and judgment. This may introduce model preference bias, as prior studies have shown that LLM judges tend to favor their own outputs or those from similar architectures.

3. The comparison with baselines may be unfair. The cold-start phase essentially distills GPT-OSS-120B into a smaller base model, while other baselines do not benefit from such distilled supervision.

4. The results lack statistical analysis. It is unclear whether the reported improvements are statistically significant or robust across runs.

5. The description of human annotation is vague; there’s no information on annotator background, quality control, or consistency checks.

6. Some implementations lack sufficient details. For instance, the loss hyperparameters are said to be chosen “based on preliminary experiments,” but no related results are shown. Similarly, defining a 0.5 score difference between human and LLM judges as “reliable” lacks empirical support.

**Questions:**

1. In the computation of 𝑅_node, since there’s no constraint on the number of entities extracted from the generated response, wouldn’t adding more entities always increase the reward?

2. How do you handle polysemy and linguistic variation in medical terminology? Could different wordings for the same concept affect the reward calculation? especially for R_struct and R_chain

3. In Figure 4, the threshold of < 0.5 is described as “reliability,” which seems arbitrary and not statistically grounded. Moreover, the boxplots look unusual, the median lines coincide with the quartile boundaries, and all four metrics show almost the same distributions. Why is that?

4. How do you define in-distribution vs. out-of-distribution data? Were the baseline models trained on these datasets, or did you train them separately? Are the reported results from the hard subset or the entire dataset?

5. The anonymous GitHub link provided in the supplementary materials appears inaccessible.

---

### Official Review · Reviewer_SghD · 2025-11-02

**Soundness:** 3
**Presentation:** 4
**Contribution:** 2
**Rating:** 2
**Confidence:** 5

**Summary:**

This paper introduces MedCEG, a framework designed to improve the clinical reliability of reasoning in large language models for medical applications. The authors argue that although reinforcement learning has improved medical reasoning performance, existing methods do not sufficiently ensure the clinical validity of generated reasoning steps. To address this, MedCEG supervises the reasoning chain using a Critical Evidence Graph (CEG), an algorithmically constructed structure representing verifiable clinical reasoning for challenging medical cases. Alongside CEG supervision, the authors develop a Clinical Reasoning Procedure Reward that evaluates the quality of reasoning based on node coverage, structural correctness, and chain completeness. Experiments across multiple medical benchmarks show that MedCEG matches or exceeds prior methods in task performance while producing more clinically accurate and interpretable reasoning chains. Overall, the work aims to enhance trustworthiness and safety in medical LLM reasoning.

**Strengths:**

* The authors construct and release a 10K clinical case dataset with structured reasoning graphs, likely valuable for future research on explainable medical LLMs.

**Weaknesses:**

**Major limitation: mischaracterization of SOTA  performance of prior work**: The Results section is missing crucial studies that mislead the reader to believe the contribution of this work achieves SOTA performance when it does not. Whether this is intentional or not, it needs to be corrected. For instance, for each benchmark, it is unclear why the best-performing models were not reported. In MedQA, GPT-4 achieves 90% accuracy, while MedGemini reaches 91%. However, the best performance reported in this work is only 75.41%. This represents a 15% gap between the current state-of-the-art and the contribution of this work. Therefore, the claim that this method “achieves SOTA performance with 58.59 on in-distribution tasks and 64.09 on out-of-distribution tasks” is incorrect. The method is far from achieving state-of-the-art performance. Please correct all tables to include SOTA performance for each benchmark

Other concerns:

1. Heavy reliance on LLMs to build supervision: CEGs and triplets are partly generated/extracted by large models, raising questions about noise, circular supervision, and bias propagation. For example, if the best average performance models can get on the evaluated benchmarks is 64% how can authors guarantee that the dataset is not correct at most 64% of the time?


2. Limited human expert validation:  The paper does not clearly quantify clinician involvement or error-rate analysis in graph construction, **critical for the medical reliability claims**. For example, prior work has extensively shown that LLMs can output the correct answer while providing incorrect clinical reasoning traces. Given that this dataset is both generated and evaluated by LLMs, how can the authors ensure its quality? I believe it is not possible to make such claims without clinical involvement

3. Evaluation focuses on QA benchmarks: Reasoning quality is mostly assessed via internal metrics and QA outcomes. This paper needs stronger real-world or expert-rated studies if claims of "produces more clinically sound reasoning chains, showing significant improvements in reasoning quality compared to existing methods" are made.

4. Missing ablation clarity: It would be helpful for the readers to see deeper ablations isolating contributions offered in the pipeline. The system looks over-engineered, but motivation is not provided

**Questions:**

Concerns are shared in above ^

---

### Official Review · Reviewer_b1Wx · 2025-11-02

**Soundness:** 3
**Presentation:** 3
**Contribution:** 3
**Rating:** 4
**Confidence:** 5

**Summary:**

The authors proposed MedCEG, a framework that augments medical models with clinically valid reasoning pathways by explicitly supervising the reasoning process through a Critical Evidence Graph. They curated a dataset and proposed a graph-based process reward function.

**Strengths:**

The authors proposed a graph-based reward function to guide the reasoning process. The reward design is based on the basic concept of graph, which makes sense.

**Weaknesses:**

1. The authors described their work as verifiable medical reasoning, but this concept is not clear to me. From what I understand, the method aligns the LLM’s reasoning process with the Critical Evidence Graph (CEG). However, there is no guarantee that the CEG itself is correct or based on accurate medical knowledge. In my view, verifiable medical reasoning should mean that the model reasons based on verified external knowledge sources. Since the model does not interact with any reliable knowledge base, I don’t think this can be considered true verifiable medical reasoning.
2. I’m very interested in the graph-based reward design. I’m particularly curious about the relationship among the node reward, structural reward, and chain reward mentioned in Formula (4). Could the authors explain how these parameters are defined? From the appendix, I noticed that the node weight is 0.5, while the structural and chain rewards are 0.3 and 0.2, respectively. If reasoning chains are important, why does the node reward receive the highest weight?
3. It also seems that the overall reward is mainly optimized by the answer itself (with a weight of 0.6). Could the authors provide additional evidence to show the effectiveness of the reasoning reward?
4. Could the authors compare the nodes in the CEG with the medical concepts mentioned in the question? I’m curious why the node features are so important in the results—especially in Figure 5. Is it because the generated CEG includes more medical concepts than the original question?
5. Is the backbone LLM used in this work LLaMA 3.1-8B?
6. The authors mention that the cold-start version is obtained by fully fine-tuning the model. Is full fine-tuning necessary? Also, why does the fully fine-tuned model perform worse than the SFT model in the in-distribution setting?
7. Could the authors compare their model’s performance with a larger model, such as GPT-OSS-20B?

**Questions:**

see weaknesses

---

### Official Review · Reviewer_YfUH · 2025-11-02

**Soundness:** 2
**Presentation:** 2
**Contribution:** 2
**Rating:** 4
**Confidence:** 4

**Summary:**

Summary
MedCEG is a graph-guided RL framework to improve verifiable clinical reasoning in medical QA. It converts rationales into Evidence Graphs (EGs) via multi-LLM triplet extraction, derives a Critical Evidence Graph (CEG) per case by backward traversal from the conclusion and transitive reduction, and trains in two stages: Cold-Start SFT using EG-linearized step-by-step text, followed by GRPO with a composite Clinical Reasoning Procedure (CRP) reward. The CRP reward evaluates Node Coverage, Structural Correctness, and Chain Completeness of the generated reasoning, alongside answer and format signals. MedCEG achieves strong results across MedQA, MedBullets-5op, MedCase and OOD benchmarks (MMLU-H, MMLU-Pro-H, DiagArena), with ablations showing each CRP component matters and CEG-based rewards outperform EG-based ones. A committee LLM process evaluation and a small human consistency check suggest improved logical coherence and evidence faithfulness.

**Strengths:**

- Novel use of CEG (minimal, causally connected subgraph) as a direct, non-learned reward signal.
- Useful composite process reward (node/structure/chain) with informative ablations.
- Release of a CEG dataset may benefit the community.

**Weaknesses:**

- Heavy reliance on LLMs for EG/CEG construction, rationale parsing, answer judging, and process scoring risks bias/circularity.
- Possible train–test overlap: training corpus is curated from benchmarks also used for evaluation; deduplication details are limited.
- Reward sensitivity: semantic matching thresholds and embedding-based node coverage may over-credit near-synonyms; limited analysis.
- Chain Completeness computed on an undirected graph; directed path validity may be more appropriate for causality.
- Evaluation leans on LLM judges; human agreement (73.6% ±0.5) is encouraging but modest, with scant details on annotator protocol/IRR.

**Questions:**

- How is train–test leakage prevented across MedQA/MedCase/JAMA? Please provide a rigorous deduplication and overlap analysis.
- What semantic similarity thresholds are used for entity/relation matching in R_node/R_struct? Include sensitivity analyses.
- Why use undirected connectivity for Chain Completeness? Can you compare against directed path coverage metrics?
- Can you add blinded human evaluation for open-ended answer correctness and report agreement with GPT-OSS judges?
- The EG-based reward “collapse” may be due to graph noise; have you tried pruned/clinician-verified EGs or hop-limited EG rewards?
- How do results vary with base model size/specialty, and does CEG complexity correlate with gains?
- Can you quantify reductions in hallucinations and calibration improvements, and provide error analyses illustrating changed failure modes?

---

### Note · Authors · 2025-11-20

I have read and agree with the venue's withdrawal policy on behalf of myself and my co-authors.